# eQTLs identify regulatory networks and drivers of variation in the individual response to sepsis

## Graphical abstract

## Authors

Katie L. Burnham, Nikhil Milind,
Wanseon Lee, ..., Charles J. Hinds,
Julian C. Knight, Emma E. Davenport

## Correspondence

julian@well.ox.ac.uk (J.C.K.),
ed5@sanger.ac.uk (E.E.D.)

## In brief

Sepsis is a life-threatening condition caused by a dysregulated host response to infection. Burnham et al. mapped genetic variants associated with differential gene regulation during sepsis and, from these, identified key regulators of a poor-outcome subgroup. This could inform targeted treatment based on the individual immune response to sepsis.

## Highlights

- Variation in the sepsis response must be resolved for treatments to be developed

- Host genetics are associated with the individual transcriptomic response to sepsis

- Identification of regulatory networks in the sepsis context pinpoints key regulators

Burnham et al., 2024, Cell Genomics 4, 100587
July 10, 2024 © 2024 The Authors. Published by Elsevier Inc.

# Cell Genomics

## Article

# eQTLs identify regulatory networks and drivers of variation in the individual response to sepsis

Katie L. Burnham,[1] Nikhil Milind,[1,2] Wanseon Lee,[1] Andrew J. Kwok,[3] Kiki Cano-Gamez,[1,3] Yuxin Mi,[3] Cyndi G. Geoghegan,[3] Ping Zhang,[3,4] GAinS Investigators, Stuart McKechnie,[5] Nicole Soranzo,[1] Charles J. Hinds,[6] Julian C. Knight,[3,4,*] and Emma E. Davenport[1,7,*]

[1]Wellcome Sanger Institute, Wellcome Genome Campus, Hinxton, UK
[2]University of Cambridge, Cambridge, UK
[3]Centre for Human Genetics, University of Oxford, Oxford, UK
[4]Chinese Academy of Medical Science Oxford Institute, University of Oxford, Oxford, UK
[5]Oxford University Hospitals NHS Foundation Trust, Oxford, UK
[6]Centre for Translational Medicine & Therapeutics, William Harvey Research Institute, Faculty of Medicine & Dentistry, Queen Mary University of London, London, UK
[7]Lead contact
*Correspondence: julian@well.ox.ac.uk (J.C.K.), ed5@sanger.ac.uk (E.E.D.)

## SUMMARY

Sepsis is a clinical syndrome of life-threatening organ dysfunction caused by a dysregulated response to infection, for which disease heterogeneity is a major obstacle to developing targeted treatments. We have previously identified gene-expression-based patient subgroups (sepsis response signatures [SRS]) informative for outcome and underlying pathophysiology. Here, we aimed to investigate the role of genetic variation in determining the host transcriptomic response and to delineate regulatory networks underlying SRS. Using genotyping and RNA-sequencing data on 638 adult sepsis patients, we report 16,049 independent expression (eQTLs) and 32 co-expression module (modQTLs) quantitative trait loci in this disease context. We identified significant interactions between SRS and genotype for 1,578 SNP-gene pairs and combined transcription factor (TF) binding site information (SNP2TFBS) and predicted regulon activity (DoRothEA) to identify candidate upstream regulators. Overall, these approaches identified putative mechanistic links between host genetic variation, cell subtypes, and the individual transcriptomic response to infection.

## INTRODUCTION

Sepsis is a clinical syndrome defined as organ dysfunction resulting from a dysregulated immune response to infection,[1] which causes an estimated 11 million deaths per year worldwide.[2] The sepsis response involves concurrent proinflammatory and immunosuppressive mechanisms, the extent and impact of which vary considerably both between and within patients over time.[3,4] The complexity and heterogeneity of the host sepsis response have limited attempts to develop targeted treatments,[5] which require better understanding of individual responses to infection and the predominant underlying mechanisms that drive organ dysfunction.[6] We, and others, have stratified sepsis patients by clinical and molecular measures to define more homogeneous subgroups.[7–11] These subgroups are hypothesized to have some distinct underlying pathophysiological mechanisms and thus could be leveraged to identify the key drivers acting in these different contexts. For example, we have previously shown that sepsis response signature (SRS) subgroups resolve the majority of transcriptomic variation in sepsis,[7] even accounting for different infection sources.[12–14] We find that SRS1 consistently identifies patients who have an immunocompromised gene expression profile and higher mor-

tality rates compared to the relatively immunocompetent SRS2, with evidence that this is driven by underlying neutrophil dysfunction and altered granulopoiesis.[14] We have developed a machine learning framework (SepstratifieR[13]) to assign SRS status based on the expression of a small set of marker genes, which can be measured with a range of technologies. Acknowledging that these subgroups likely capture different ranges of a continuously varying trait, we additionally developed a quantitative score, SRSq,[13] which encompasses a spectrum from health (values close to 0) through SRS2 to SRS1 (values close to 1). However, the regulatory determinants and predisposing factors for the high-risk SRS1 state are unclear.

Host genetic background is a plausible driver of variation in the response to infection,[15–18] and in this regard, the use of genome-wide association studies (GWASs) and rare variant analysis has proved highly informative in severe COVID-19.[19–23] While these approaches have also been applied in sepsis,[16,17,24–26] they have been less successful, again likely due to disease heterogeneity and smaller sample sizes. We hypothesize that the host response to sepsis, as represented by the transcriptomic SRS subgroups, has a polygenic basis with additional environmental modulators and that elucidation of these underlying regulatory networks will be key in understanding interindividual variation in

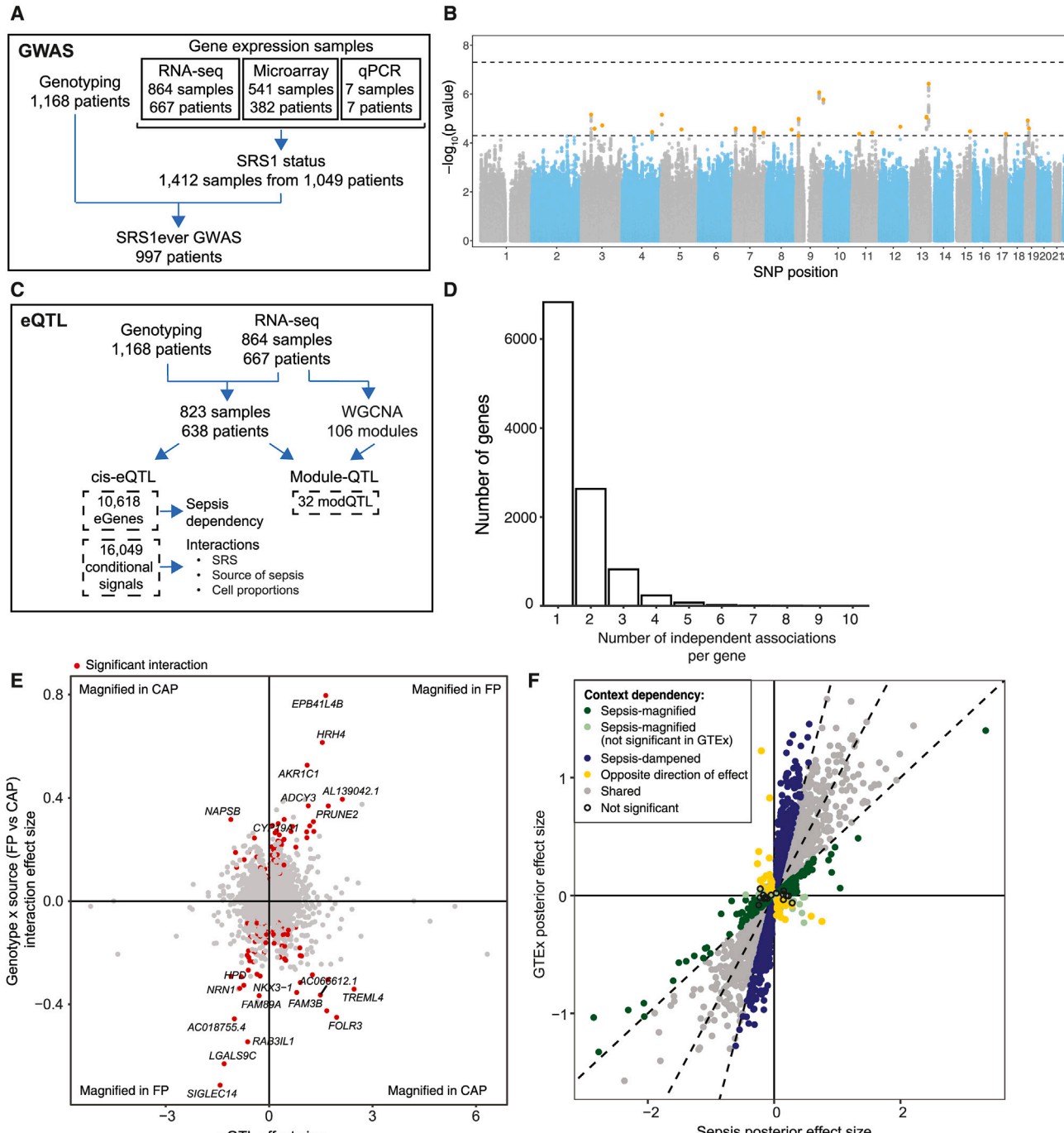

**Figure 1. Genetic variants associated with gene expression in the context of sepsis**

(A) Schematic of cohort design for SRS1ever genome-wide association study (GWAS) using all patients with genotyping data and at least one gene-expression time point for SRS assignment.

(B) Common SNPs (MAF ≥ 1%) were tested for association with the SRS1ever vs. never phenotype. Manhattan plot showing −log10($p$ value) for each variant plotted against its genomic position. The most significant SNP in each locus is highlighted in orange.

(C) Schematic of cohort design for eQTL (cis-eQTL and co-expression module QTL) analysis using all samples from patients with genotyping data and RNA-seq data. Co-expression modules were defined using the full RNA-seq dataset.

(D) Histogram showing the distribution of the numbers of independent signals detected through conditional analysis for each eGene.

disease responses and identifying treatable traits.[27] Genetic factors, notably non-coding variants, may modulate gene expression as expression quantitative trait loci (eQTLs). Importantly, such associations can vary across cell types and states,[28–30] meaning that it is vital to map eQTLs in the context of disease to understand the contribution of regulatory variants to pathogenesis.[31] In addition, specific instances of transcriptomic regulation may be affected by external factors, resulting in a genotype-by-environment interaction. Identification of a set of eQTLs that are modulated by the same environmental variable can therefore reveal important upstream regulators and pathways through which the perturbation impacts biological function.[28,32]

Here, we apply this approach to sepsis, particularly investigating the influence of genetic variation on transcriptomic subgroups that may represent divergent regulatory networks. The UK Genomic Advances in Sepsis (GAinS) study has recruited a cohort of >1,400 patients with sepsis due to community-acquired pneumonia (CAP) and fecal peritonitis (FP), admitted to adult intensive care units (ICUs) across the United Kingdom. We have generated array genotyping data on 1,168 patients and genome-wide gene expression data on 1,043 unique patients.[7,12,13] This provides an opportunity to identify genetic drivers of gene expression in a disease context with high power to detect interaction effects related to patient phenotypes. We find evidence of genetic variation associated with widespread transcriptomic differences in the sepsis response and leverage this to identify putative key regulators of SRS.

## RESULTS

### Evidence for contribution of genetic variation to SRS

We have assigned SRS group membership to 997 GAinS sepsis patients with both genotyping and blood gene expression data (from qPCR, microarray, or RNA sequencing [RNA-seq][7,12,13]) collected at multiple time points during the first 5 days of ICU admission (Figure 1A; Table S1). We first aimed to estimate the overall contribution of common genetic variation to the SRS phenotype using paired genotyping and, as SRS status for a given patient may change over time with disease natural history, whether a patient had ever been assigned to SRS1 during the period of observation. We estimated the heritability[33] as 57% (±28%, $p = 0.019$, $n = 440$ "SRS1ever" vs. 557 "SRS1never" patients), supporting the hypothesis that common variants contribute substantially to SRS during critical illness (Figure S1; Table S2).

We proceeded to carry out a GWAS for SRS1ever vs. never. This identified 155 variants in 25 loci reaching the genome-wide suggestive $p$-value threshold of $5 \times 10^{-5}$, henceforth referred to as "SRS1 GWAS SNPs" (Figures 1B and S2; Table S3). These include SNPs previously reported to be associated with monocyte, lymphocyte, and platelet counts; clotting factors; and lung function[34,35] (Table S4).

### eQTLs in sepsis patients

To identify genetic regulators of gene expression, we then mapped *cis*-eQTLs using a random intercept linear mixed model[28] (823 samples from 638 sepsis patients with bulk RNA-seq and genotyping, Figures 1C and S3; STAR Methods) and identified significant associations for 10,618 genes (eGenes) (Table S5). Given the importance of the major histocompatibility complex (MHC) region in immunity,[36] we ensured accurate quantification of human leukocyte antigen (HLA) gene expression and eQTL identification through remapping to personalized references (STAR Methods). Through conditional analysis we found 16,049 independent associations in total (Table S6), with multiple signals for 3,788 eGenes (35.7% of all eGenes, median of 1 and maximum of 10 signals for each gene) (Figure 1D). As previously reported in healthy cohort studies,[37] we observed that the expression-associated SNPs (eSNPs) underlying primary signals were more common and located closer to the transcription start site (TSS) of the associated gene than were secondary/tertiary+ signal variants (Figure S4). Five eQTLs significant in sepsis patients involved SRS1 GWAS SNPs (Table S4). Of these, two eQTL signals in the same region (*OCEL1*, a predicted membrane component of unknown function,[38] and *NR2F6*, an immune checkpoint that suppresses adaptive responses[39]) showed evidence of colocalization[40] with the same GWAS locus (posterior probability for a single variant affecting both traits [PP4] 90.3% and 89.9%, respectively), indicating that the same causal variant is driving these signals (Figure S5).

We have previously described eQTLs in a smaller microarray dataset of patients with sepsis due to CAP (*n* = 240, of which 134 individuals overlap with the RNA-seq cohort, STAR Methods).[7] After restricting the new RNA-seq dataset to the 689 non-overlapping samples, we found high correlation of eQTL effect sizes with those previously reported across SNP-gene pairs assayed in both datasets (Pearson's r = 0.70, Figure S6; Table S7). However, in the full RNA-seq cohort we also identified more than twice the number of eGenes, likely due to the larger sample size and the greater sensitivity of RNA-seq for detecting expression. This expanded cohort also encompasses a broader range of patients, including abdominal sepsis, meaning some of the additional eQTLs detected here may be context dependent. We tested each of the independently associated lead SNPs for interaction effects with the source of sepsis (CAP or FP, 12,663 SNP-gene pairs with ≥2 minor allele

(E) eQTL interactions with source of sepsis (CAP or FP). Each point represents an independent eSNP-eGene pair, with the interaction effect size plotted against the genotype effect. eQTLs with bigger effects in FP compared to CAP are therefore found in the top right and bottom left quadrants. Red indicates a significant interaction between genotype and source of sepsis (FDR < 0.05), with the most significant results labeled with the eGene name.

(F) Sepsis-dependent eQTL effects identified with mashr. Each point represents a lead SNP-eGene pair from the first-pass eQTL mapping in sepsis patients that was also tested for whole-blood eQTL in the European subset of GTEx. Posterior effect sizes estimated by mashr are plotted for GTEx against sepsis, and eQTLs are categorized based on the difference between these estimates. eQTLs significant in sepsis are "shared" if the mashr posterior effect size is in the same direction as and within a factor of 0.5 of the GTEx effect size. Those with bigger effects in the same direction in sepsis or GTEx are "sepsis-magnified" and "sepsis-dampened," respectively. Those significant in both GAinS and GTEx but with opposite directions of effects are "opposite direction of effect" (*n* = 53). Those significant only in sepsis are also classed as "sepsis-magnified," and those significant in neither cohort are "not significant." Please see also Figures S1–S9 and Tables S2, S3, S4, S5, S6, S7, S8, S9, and S10.

homozygotes in each group tested, STAR Methods). We identified 166 significant interaction effects (false discovery rate [FDR] < 0.05), more than expected by chance (permutation $p < 0.01$, Figure S7; Table S8), of which roughly half ($n = 88$) had stronger effects in CAP (Figure 1E). The eGenes involved were enriched for the Reactome term "biological oxidations" (FDR = 0.0033) and were members of a subnetwork connected by the hub genes *APP*, *AKT1*, and *ABCC1* (STAR Methods). For comparison and given the known association between sex and infectious disease,[41–43] we similarly tested for autosomal eQTL interactions with sex and found only nine significant effects (Table S9).

To explore sepsis dependency of observed eQTLs, we used Genotype-Tissue Expression (GTEx) and eQTLgen, two well-characterized blood eQTL datasets from healthy donors.[44,45] As expected, the majority of effects were highly correlated between health and sepsis (Pearson's r = 0.898, 0.803) but we did find *cis*-eQTLs in sepsis not detected in either healthy dataset (38.7% and 17% of our lead SNP-gene pairs were called nonsignificant in GTEx and eQTLgen, respectively) (Figure S8). Given this evidence, we further quantified this context dependency by comparing eQTL effect sizes significant in sepsis to GTEx, a similarly sized bulk RNA-seq cohort, with mashr[46] (STAR Methods). We categorized our eQTLs significant in sepsis as "shared" ($n = 5,943$) or "context dependent" ($n = 2,179$, Figure 1F). Specifically, we identified 854 signals with bigger and/ or only significant effects in sepsis ("sepsis magnified"), 1,272 with bigger effects in healthy volunteers ("sepsis dampened"), and 53 eQTLs that were significant in both GAinS and GTEx but had opposite directions of effect (Table S10). We found no significantly enriched Reactome pathways within these eGene subsets. However, sepsis-magnified eQTLs differ significantly from those shared with GTEx, with the eSNPs involved having a lower minor-allele frequency (MAF) (median 11.5% vs. 12%, Mann-Whitney $p = 0.00015$) and being farther away from the TSS of their target gene (median |distance| = 52.1 vs. 27.6 kb, Mann-Whitney $p = 2.5e{-}27$, Figure S9).

### eQTL effects vary between sepsis patients

We proceeded to investigate the distinct regulatory networks underlying the SRS groups by identifying differential genetic regulation of gene expression between samples assigned to SRS1 and non-SRS1. We found a significant interaction between genotype and SRS status in 1,578/12,959 of SNP-gene pairs (12%), with $^2/_3$ of these having a more pronounced effect on gene expression in SRS1 (1,064 magnified in SRS1 vs. 514 dampened; Figures 2A and S10; Table S11). eGenes with a magnifying SRS interaction were enriched for the Reactome term "synthesis of PA [phosphatidic acid]" (FDR = 0.0069). We have previously noted the relevance of metabolic changes and glycolysis in SRS1,[7] so it is possible that PA is relevant in this respect, given previous evidence of a role in mTORC1 activation and in regulation of systemic inflammation.[48–50] All SRS interaction eGenes were enriched for genes differentially expressed between SRS groups[13] (Fisher's exact test [FET] $p = 4.59 \times 10^{-4}$), with 628 magnifiers and 105 dampeners involving genes that were significantly upregulated in SRS1 (enrichment of SRS1 magnifiers in upregulated genes FET $p = 5.3 \times 10^{-63}$). We noted

that sepsis-magnified eGenes were significantly enriched for SRS interactions (FET $p = 0.03$), indicating that genetic regulation of some genes may be modulated on a spectrum from health to SRS2 to SRS1. This is illustrated by a subset of sepsis-magnified eGenes where the effect of genotype also increases continuously with SRSq score, a quantitative measure for SRS,[13] for example, *FAM89A*, a gene previously highlighted as a potential biomarker for pediatric bacterial infection[51] (Figure 2B).

As previously demonstrated in healthy cohorts,[52,53] we identified many eQTLs with putative cell-type-specific effects by adding genotype-cell proportion interaction terms to our eQTL model. Specifically, we found 1,073, 1,013, and 608 eQTLs with significant interactions (FDR < 0.05) with measured neutrophil, lymphocyte, and monocyte proportions, respectively (Figure S11; Table S12). The larger number of interactions detected in neutrophils and lymphocytes may be due to those cell types being more prevalent and having greater variance across sepsis patients. As cell proportions are necessarily related, we observed highly correlated and reciprocal effects across interaction types; for example, eQTLs whose effect is amplified in the context of high neutrophil proportions ("neutrophil magnifiers") generally also had a larger effect size with lower lymphocyte proportions ("lymphocyte dampeners") (Figure 2C).

We observed that SRS interaction eSNPs were enriched (one-tailed binomial test FDR < 0.05) in neutrophil and monocyte enhancers and flanking TSS regions across primary immune cells[47] (Figure 2D; STAR Methods), indicating the particular contribution of myeloid cells to the transcriptomic differences between SRS groups. Given recent findings that disrupted granulopoiesis is a key feature of the SRS1 subgroup,[14] we were interested to note that SRS1 magnifiers showed a large overlap with neutrophil magnifiers (496/1,064) (Figure 2C). One such SRS and neutrophil magnifier eQTL, associated with reduced expression of *OCEL1*, involved an SRS GWAS SNP (rs891204).

### Upstream regulatory drivers of SRS

We then used the SRS interaction eQTLs to identify putative upstream drivers of the differential host response to sepsis. We hypothesized that variation in the impact of regulatory variants could be mediated by changes in the activity of a transcription factor (TF) whose binding is affected by the eSNPs. We therefore first identified instances where each of 124 human TF binding motifs was interrupted or introduced by eSNPs for eQTLs significant in sepsis (or their linkage disequilibrium [LD] proxies) using SNP2TFBS[54,55] (Figure 3A). We found this was more common when the eQTL had an SRS interaction (median 5 vs. 4 binding sites introduced/interrupted per eGene, Wilcoxon $p = 1.08 \times 10^{-6}$). For each TF motif, we then classified eGenes as having at least one or no binding sites altered. We found that 56 TF motifs were significantly enriched for alteration among SRS interaction eGenes (Figure 3B; Table S13), with the HIF1A-ARNT motif having greatest enrichment. This was significantly more motifs than expected by chance ($p < 0.001$), as computed by permuting eGene interaction status 1,000 times and repeating the enrichment analysis (Figure S12).

However, these predicted binding sites may not be occupied *in vivo*, and experimental binding data are currently limited to small numbers of factors and contexts. We therefore calculated

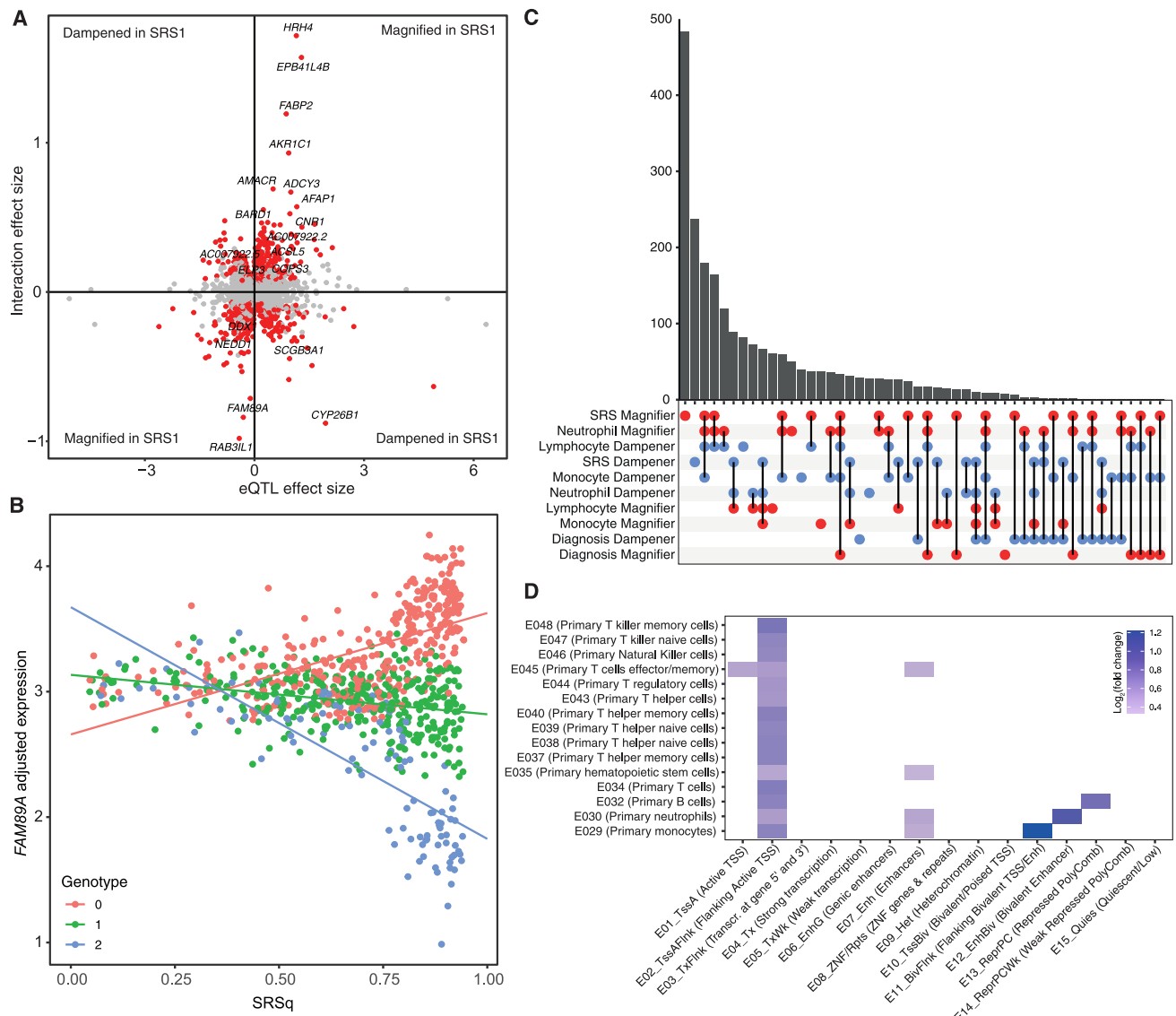

**Figure 2. Genotype-by-environment interactions find widespread variation in eQTL effects across sepsis patients**

(A) eQTL interactions with SRS1 status. Each point represents an independent eSNP-eGene pair, with SRS interaction effect size plotted against the genotype effect. eQTLs with bigger effects in SRS1 compared to non-SRS1 are therefore found in the top right and bottom left quadrants. Red indicates a significant interaction between genotype and SRS1 status (FDR < 0.05), with the most significant results labeled with the eGene name.

(B) An exemplar sepsis-magnified eQTL that also has a significant positive interaction with SRS1 status. Gene expression residuals were modeled with an SRSq-by-genotype interaction to illustrate the continuous relationship of SRS status with the genotype effect, with point color indicating number of copies of the minor allele of rs4378192.

(C) UpSet plot showing sharing of magnifying (red) and dampening (blue) eQTL interaction effects between environmental variables tested.

(D) Enrichment of eSNPs for eQTLs with an SRS interaction in different chromatin states from the Roadmap Epigenomics core 15-state genome segmentation annotations[47] across relevant cell types, compared to eSNPs for eQTLs without a significant SRS interaction (only significant enrichments shown [FDR < 0.05]). Please see also Figures S10 and S11 and Tables S11 and S12.

TF activity scores in each sample using 288 curated regulons from DoRothEA to pinpoint regulators predicted to vary by SRS (STAR Methods).[56,57] We found that 253/288 factors had significantly differential inferred activity between SRS groups (Figure 3C; Table S14). Given the previously reported extensive differential gene expression between SRS groups,[7,13] this is not surprising, but we confirmed that it was more than expected

by chance by permuting SRS status and recalculating differential activity ($p < 0.001$, Figure S13).

Of the TFs with differential activity, 43 were also enriched in the SNP2TFBS analysis (40 motifs), indicating that they could be driving the differences in the regulatory landscape between SRS groups (Figure 3D). These included BATF-JUN, ZNF263, HIF1A-ARNT, CEBPB, SPIB, NFE2-MAF, and FOSL1. We also utilized

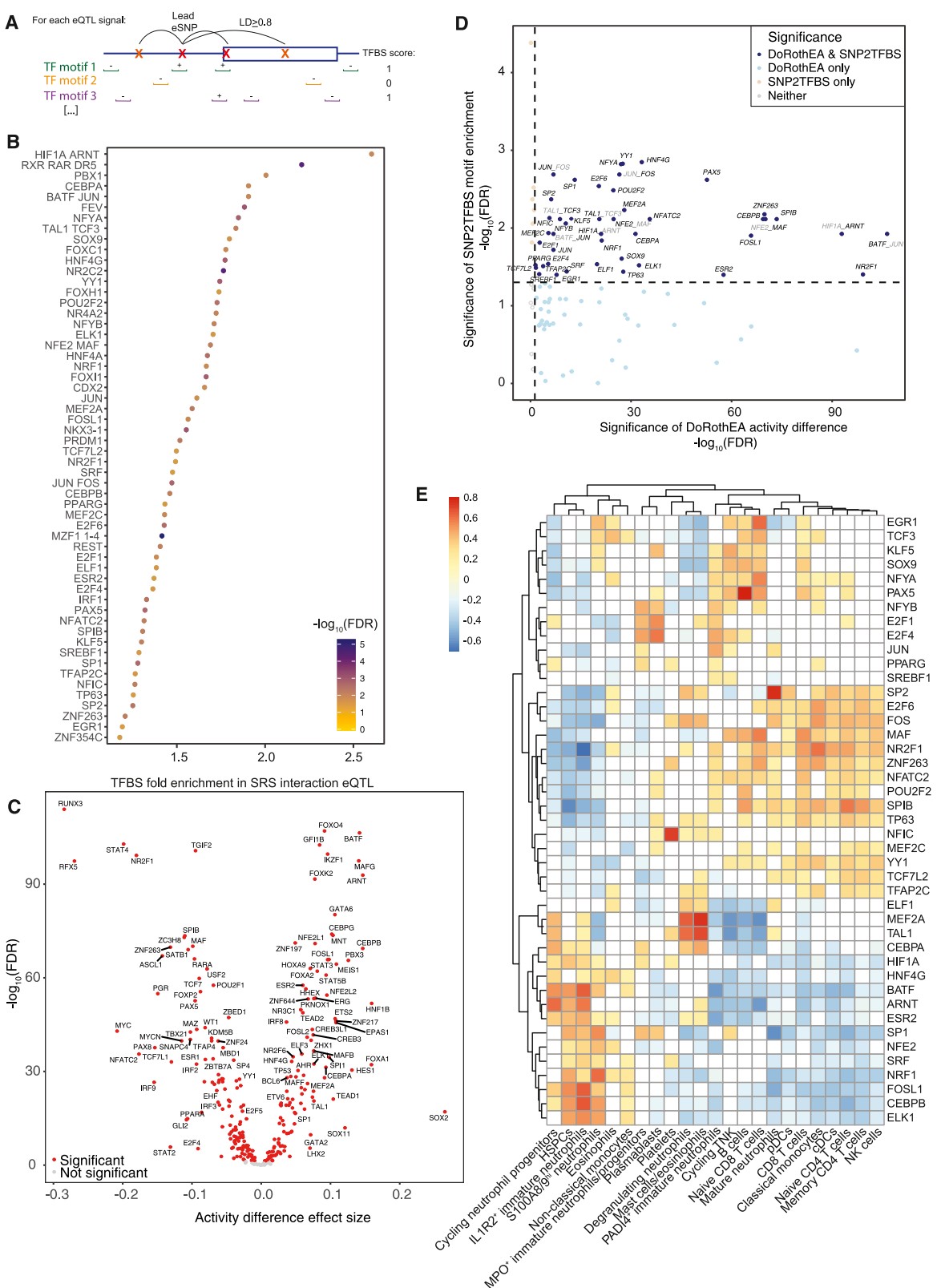

(legend on next page)

cell proportions estimated in each sample using a sepsis single-cell reference dataset[14] to identify relationships between these TFs and 23 specific cell types. The inferred activity of many of the putative driver TFs was weakly correlated with a number of the 23 estimated cell subsets, but some had strong and specific associations, potentially indicating cell-type specificity. These included PAX5 with B cells (rho = 0.80); SP2 with mature neutrophils (rho = 0.76); MEF2A and TAL1 with mast cells/eosinophils (rho = 0.75 and 0.69) and degranulating neutrophils (0.64 and 0.61); NFIC with platelets (0.72); CEBPB, FOSL1, and BATF with IL1R2$^+$ immature neutrophils (0.66, 0.65, and 0.64); and ARNT with cycling neutrophil progenitors (0.62) (Figure 3E; Table S15).

## Co-expression module QTL mapping

Finally, we aimed to investigate the regulation of gene expression in *trans*. Given our cohort size, we focused on one possible mechanism where a single SNP is associated with the expression of multiple related distal genes via a shared upstream regulator.[45] To increase our power to detect such *trans*-regulatory networks, we leveraged metrics summarizing the expression patterns of co-expressed gene sets, rather than testing each gene individually.

Using the WGCNA package,[58,59] we identified 106 co-expression modules, each comprising 11–1,785 genes (Table S16; Figure S14) with highly correlated gene expression. We summarized the expression of each module with its primary eigengene (the first principal component) to provide a single representative value for the module in each sample. We then correlated these module eigengenes with our features of interest and found that individual modules were associated with disease phenotypes, including SRS, and survival and measured cell proportions (Figure S15; Table S17). Furthermore, these modules were enriched for biological pathways and marker genes for more granular blood cell populations,[14,60] suggesting they capture gene sets associated with specific biological processes and/or cell types (Figures 4A and S16; Tables S18, S19, and S20). Finally, to investigate whether these co-expression modules represent sets of co-regulated genes, we tested the set of genes in each module for enrichment of known TF targets from DoRothEA[56] and found at least one significant TF for 43/106 modules (Table S21).

To identify genetic regulators of the transcriptomic programs captured by these modules, we tested for association between each module eigengene and the lead *cis*-eSNPs identified here with >3 minor allele homozygotes in the cohort (12,335 SNPs). We found 241 significant associations for 30 modules across

32 loci ($p < 0.05/(12,335 \times 106) = 3.82^{-8}$), which we termed module QTL (modQTL) (Table S22). We then sought to replicate these modQTLs using our previously published microarray gene expression data[7,12,13] ($n = 135$ samples overlapping RNA-seq, $n = 506$ non-overlapping). We computed eigengene values for each module gene set with at least five genes measured on the microarray. Comparing the module eigengenes across technologies using the 135 overlapping samples, we found that, in general, the correlation was good but varied by module (median |rho| for correlation between RNA-seq and microarray module eigengene values 0.77, range 0.04–0.98). We found that 16/29 lead modQTLs replicated in non-overlapping patients (Figure S17; Table S23) and noted that replicating modQTLs had greater correlation between the RNA-seq and the microarray eigengene values computed for the samples included in both datasets (median |rho| for module eigengenes with replicating modQTLs 0.73, for non-replicating modQTLs 0.49, Figure S18).

We noted that for the majority of modQTLs (30/32) the module contained at least one *cis*-eGene for the modQTL SNPs. While this could provide support for *trans* regulation of the rest of the module genes via a *cis*-eGene, the modQTLs could also be a result of the *cis*-eGene(s) dominating the module eigengene and thus driving the overall modQTL signal. Moreover, several modQTLs involved clusters of genes close to the SNP, with few or no distal genes included in the module. Such modQTLs could be driven by a single eSNP regulating multiple genes in *cis* or haplotypes comprising multiple *cis*-eSNPs with the inferred co-regulation driven by LD between *cis*-regulatory elements for multiple nearby genes. We therefore performed a stringent sensitivity analysis to prioritize modQTLs with evidence of *trans* regulation. We recalculated the module eigengenes excluding all *cis*-eGenes associated with the modQTL SNPs in the conditional eQTL analysis and retested for association with the lead module SNP. This identified 7/32 robust modQTLs that remained significant using the reduced module gene sets (Figure S19; Table S23). We then tested for mediation[61] by the *cis*-eGene(s) of associations between the lead modQTL eSNP and the recalculated eigengene, and for all seven modQTLs, we found significant evidence of mediation by an implicated *cis*-eGene (Table S24).

Several of these prioritized modQTLs involved disease-relevant modules, where the module eigengene was also associated with clinical phenotypes (Figure 4B; Table S17), highlighting regulators that may be contributing to disease heterogeneity

**Figure 3. Identification of putative driver transcription factors for SRS from eQTL interactions**

(A) Schematic of strategy to identify transcription factor (TF) binding sites (TFBSs) enriched in SRS interaction eQTLs. For each of 12,959 eQTL signals tested for an SRS interaction effect, query SNPs were defined as those in LD ($r^2 \geq 0.8$) with the lead SNP. We then identified instances where binding motifs for 124 human TFs were interrupted or introduced by these query SNPs using SNP2TFBS. Each independent eQTL signal was scored for each motif as having $\geq 1$ or 0 binding sites altered by the signal SNP or its LD proxies. We then tested for enrichment of each TF motif among eQTLs with a significant interaction effect compared to eQTLs with no significant interaction using a one-tailed Fisher's exact test.

(B) TF binding sites identified with SNP2TFBS enriched in eQTLs with an SRS interaction vs. no interaction, with point color indicating significance.

(C) Volcano plot showing comparison of inferred TF activity between SRS1 and non-SRS1 samples. Each point represents one TF, with adjusted $p$ value plotted against effect size estimated from a linear mixed model. Red indicates significance.

(D) Adjusted $p$ values from (B) are plotted against those from (C) to highlight TFs with both enriched binding sites in SRS interaction eQTLs and differential activity between SRSs.

(E) Heatmap showing Spearman correlation between estimated cell proportions and inferred TF activity for the first sample available per patient. White indicates a non-significant correlation (FDR > 0.05), red a positive correlation, and blue a negative correlation. Please see also Figures S12 and S13 and Tables S13, S14, and S15.

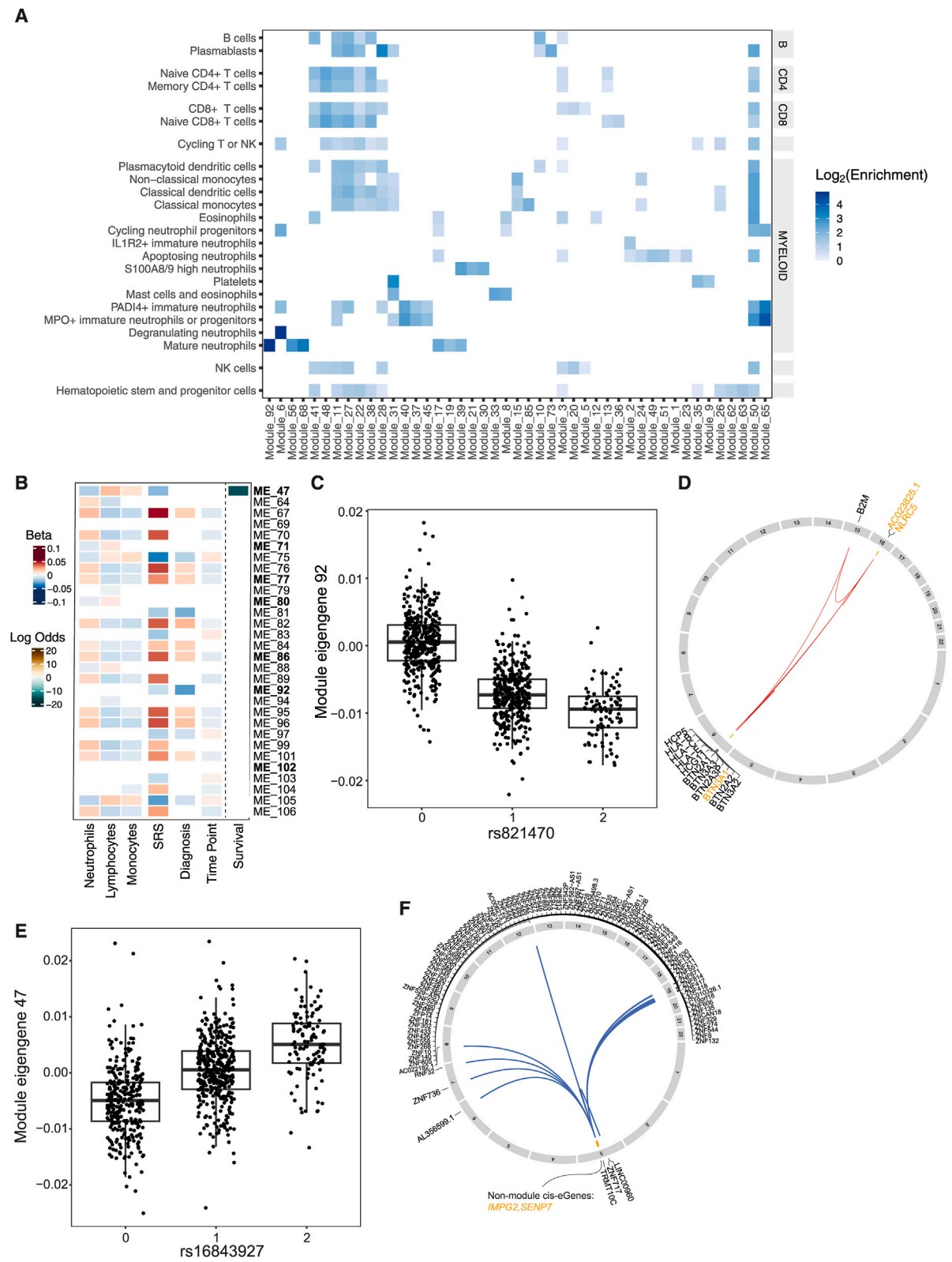

(legend on next page)

through modulation of a gene network. For example, we identified a replicating modQTL for module 92 (Figure 4C), which was associated with SRS and diagnosis and enriched for mature neutrophil marker genes[14] (Figure 4A). It also included several HLA class I genes, for which we have increased confidence in the gene expression quantification due to the incorporation of personalized HLA references in our RNA-seq mapping pipeline (Figure 4D). Of the two implicated *cis*-eGenes, *NLRC5*, a known regulator of HLA class I genes with a key role in inflammatory processes,[62] was significantly upregulated in SRS1 and was a significant mediator of the modQTL effect (Figure S20).

We additionally identified and replicated a modQTL for module 47 (Figure 4E), which was associated with SRS1 status and mortality. The lead modQTL SNP, rs16843927, has been previously associated with inflammatory bowel disease through GWASs,[34,35,63] and we found evidence for colocalization[40] of the modQTL and GWAS signals (PP4 = 99.1%). The module primarily consists of a cluster of zinc-finger genes on chromosome 19 involved in retroviral repression and was enriched for targets of STAT1 and ZNF274, with ZNF274 itself included in the module gene set (Figure 4F). The *cis*-eGenes for the modQTL SNPs *SENP7* and *IMPG2* were not module members, but we observed significant partial mediation (*p* = 0.018) by *SENP7* of the association between the SNP and module gene expression (Figure S20). *SENP7* has previously been reported to regulate many of the module 47 zinc-finger genes in *trans*,[64] further supporting this association.

## DISCUSSION

There are currently no immunomodulatory drugs available to treat the dysregulated immune response underlying sepsis, despite a large number of initially promising candidates.[5] Drug targets with genetic evidence are at least twice as likely to be successful in clinical trials,[65,66] so leveraging genetic associations for sepsis phenotypes is an important avenue to explore. Here, we have investigated the role of genetic variation in the sepsis response, finding evidence for genetic risk factors, and identified putative driver TFs and regulatory networks underlying interindividual variation in the response to infection.

### Employing the GWAS approach in sepsis

We focused our GWAS on a composite molecular phenotype rather than an outcome such as mortality to improve power to detect genetic associations and aid mechanistic interpretation. While the lack of available expression data to assign SRS in

external cohorts limits our ability to validate our findings, we did find that one previously reported mortality-associated variant, rs72998754,[67] was in strong LD (r² = 0.98 in 1KGP EUR) with nominally significant SNPs in our SRS GWAS (*p* = 0.00015). Furthermore, by integrating our eQTL data from the sepsis context, we find several plausible drivers of variation in the individual sepsis response. For example, one locus involves an eSNP for *NR2F6*, a TF that represses transcription of key cytokines in CD4+ and CD8+ effector T cells, including IL-2, IFN-γ, and TNF-α.[68,69] The eSNP is located within an ATF3 chromatin immunoprecipitation (ChIP) peak[70] and therefore may be specifically relevant in the context of stress and immune regulation. The causative pathogen and site of infection are key sources of variation in the sepsis response, and it is likely that there are genetic risk factors specific to particular types of infection. Previous reports raise the possibility that this *NR2F6* association is particularly relevant to bacterial peritonitis,[71–73] but our limited cohort size and microbiological information preclude subgroup analyses. With methodological advances, such investigations will be an important direction for future inquiry, together with consideration of sex-specific risk factors.

### Context specificity of eQTL effects

It has been widely reported that, despite their presumed regulatory activity, GWAS associations have only limited overlap with eQTLs.[74] One possible explanation for this observation is that the majority of eQTL studies have been conducted on healthy individuals, and regulatory variants may be active only in disease-relevant conditions, such as following immune activation. As sepsis represents an extreme and systemic response to infection, our eQTL results in the context of sepsis may help interpretation of risk variants for a broad range of immune and inflammatory diseases. Moreover, elucidating how environmental context impacts regulatory associations may improve understanding of how genetic variants contribute to complex traits. In addition, we found that nearly 2,000 eQTL signals significant in sepsis had a significant interaction with SRS, source of sepsis, and/or measured cell proportions. Given that our approach required an eQTL signal in the full cohort prior to interaction testing, interactions with opposite effects may not have been detected due to the average main eQTL effect being non-significant.

### Cell proportion differences as drivers of eQTL interactions

Transcriptomic regulation is known to be highly variable across cell types.[75–77] The eQTLs detected in bulk gene expression

---

**Figure 4. Co-expression modules pinpoint several *trans*-regulatory networks relevant to sepsis outcomes**

(A) Heatmap showing enrichment of sepsis cell markers in module member genes. Modules were tested for enrichment of leukocyte marker genes identified in a sepsis cohort.[14] Modules shown had significant enrichment for at least one signature.

(B) Heatmap showing significant associations between module eigengenes (MEs) with a modQTL and clinical phenotypes. modQTLs passing stringent sensitivity analysis are marked with bold font. MEs were tested for differential expression with measured cell proportions, SRS1 status, diagnosis (CAP or FP), and time point (day 1, 3, or 5) using a linear mixed model. Association of each ME with survival up to 28 days was tested using a Cox proportional hazards model.

(C) Module 92 eigengene (ME_92) plotted by rs821470 genotype, with partial residuals calculated from the linear-mixed-model fit.

(D) Circos plot showing the chromosomal locations of the genes contained in module 92 and the lead eSNPs associated with the ME. Member genes that are *cis*-eGenes for these eSNPs are highlighted in orange.

(E) ME_47 plotted by rs16843927 genotype, with partial residuals calculated using the linear-mixed-model fit.

(F) Circos plot showing the chromosomal locations of the genes contained in module 47 and the lead eSNPs associated with the ME. Genes that are *cis*-eGenes for these eSNPs are highlighted in orange. Please see also Figures S14–S20 and Tables S16, S17, S18, S19, S20, S21, S22, S23, and S24.

from a mixed cell population could therefore be affected by varying proportions of different cell types across patients, as well as in comparison to health. Development of sepsis and sepsis severity are associated with profound changes in leukocyte proportions; in particular, expansion of the neutrophil compartment has been observed in sepsis,[14,78,79] COVID-19,[80] and acute respiratory distress syndrome (ARDS).[81] This involves both mature neutrophils, known to be increased in sepsis, and immature neutrophil subpopulations that are rare or absent in the circulation of healthy individuals but are mobilized in response to infection. It is likely that some sepsis-magnified or specific eQTLs derive from these leukocyte subpopulations. As immunosuppressive neutrophil subsets are associated with secondary infections in sepsis,[82] such effects could be of particular relevance to outcomes. However, around half of our SRS1-magnifying interactions did not overlap with a cell-type interaction, and therefore, differences in cell types do not explain all the effects observed and additionally may not pinpoint the causative drivers involved.

### Identifying putative drivers of SRS

Heterogeneity across patients is a major obstacle to the identification of risk factors and the development of targeted treatments in sepsis.[83] Identification of the hub genes that modulate networks could enable personalized, targeted immunomodulation in sepsis. We identified 43 putative driver TFs with differential activity between SRS groups and for which the binding motif is also enriched in SRS interaction QTL.

The most significantly enriched motif was for the HIF-1 TF complex comprising HIF1α and ARNT (HIF1β). We previously reported that a network centered on HIF1α was upregulated in the poor-outcome SRS1 patients,[7] and we replicate this finding here, with both HIF1α and ARNT showing greater gene expression and elevated inferred activity in SRS1. While HIF1α activation occurs in response to hypoxia and inflammatory signaling following infection, ARNT is constitutively expressed, and its regulation is less well characterized.[84,85] HIF-1 has been widely identified as important in the sepsis response, inducing metabolic reprogramming in immune cells[86] as well as affecting coagulation,[87] cell proliferation, and apoptosis,[88] with a more pronounced impact in non-survivors.[89,90] The HIF1A-ARNT motif was also enriched among neutrophil-interaction eQTLs, and we found that HIF1α activity was correlated with the estimated proportions of platelets and immature neutrophils, and ARNT was correlated with cycling neutrophil progenitors and IL1R2$^+$ immature neutrophils. When we previously characterized neutrophil subtypes in sepsis with an independent single-cell dataset, we defined an IL1R2$^+$ gene expression program correlated with SRSq that was enriched for glycolytic and hypoxia-related pathways, highlighting a specific cell state where alterations in metabolism may be particularly important.[14]

The prioritized drivers also included CEBPA and CEBPB, regulators of steady-state and emergency granulopoiesis (EG), respectively.[91] While the binding motifs tested are very similar, we previously identified CEBPB in particular as a key factor in the SRS1 response in our single-cell study,[14] and here, we find a bigger difference in activity between SRS groups for CEBPB, further supporting the importance of EG in SRS1 with

an independent approach. Furthermore, the inferred activity is correlated with immature neutrophil subsets as previously described.[14]

### modQTLs to identify genome-wide regulatory networks

Our eQTL interaction framework is one approach for identifying upstream regulators of groups of genes. We also utilized co-expression modules as a way of identifying putative *trans*-regulatory networks with relevance to sepsis pathophysiology.[92–94] This approach identified a small number of associations, largely involving smaller modules (under 40 genes). Recent work has demonstrated that eQTL genes are less connected in co-expression networks compared to nearest genes of matched SNPs and GWAS associations.[74] Genes that are well connected in gene-regulatory networks are more important and thus require tightly controlled expression, resulting in higher constraint and fewer eQTLs due to negative selection. This reduces the power to discover *cis*-eQTLs for genes involved in large regulatory networks. Thus, our module analysis was likely powered to detect modQTLs for genes involved in smaller regulatory networks.

Although modules in bulk expression cannot capture the cell-type specificity of gene-regulatory networks in a heterogeneous tissue like blood,[95] we were able to compare our gene sets with marker genes derived from single-cell RNA-seq in a smaller subset of individuals with sepsis.[14] This revealed that some of our modules captured cell-type signatures that are specific to immune cells in sepsis. Beyond the ability to generate data at scale, an advantage of identifying these signatures in bulk data is that they are more amenable to translation into a clinical setting. Of the 32 modQTLs that we identified, module 47 was particularly interesting, with associations with SRS and mortality and module members including zinc-finger genes involved in retroviral repression. We have previously described increased Epstein-Barr virus (EBV) viral load in SRS1 patients.[96] It is still unclear whether this association is a pathological effect rather than simply an epiphenomenon, but this additional understanding of gene regulation indicates that further exploration is warranted.

### Limitations of the study

There are necessarily limitations to our study. We adopted this eQTL framework because of the practical limitations of collecting sufficient samples in critical illness for well-powered *trans* eQTLs and GWAS, particularly given that SRS status requires the quantification of gene expression. We generated bulk RNA-seq to achieve sufficient sample sizes for eQTL mapping, but interpretation of cell-type-specific effects is limited to those that can be deconvoluted using a single-cell reference. Given the importance of neutrophils in the response to sepsis, it is key to generate single-cell data from whole blood rather than peripheral blood mononuclear cells; however, it is still challenging to do this at scale. Technological advances will allow investigation of the more granular cell types contributing to the observed heterogeneity. Although the SRS gene expression signatures have been replicated across cohorts,[13] the lack of availability of both genotyping and gene expression on the same sepsis patients also precluded validation of the genetic associations in external cohorts given the fairly substantial sample sizes that would be required. For example, power calculations indicate that sample

sizes of 300–470 patients would be necessary for replication of the SRS GWAS results. Our cohort was restricted to the two main sources of sepsis in the United Kingdom (CAP and FP). While we observed a small number of eQTL effects differing by source of infection, potentially driven by location and/or pathogen type, we were not able to investigate pathogen-specific effects due to insufficient microbiological information. Our cohort is of predominantly European ancestry, and therefore population-specific effects will not be identified. Finally, there may be other features unique to this UK cohort, necessitating validation in external datasets. As sampling technologies and study design evolve, this may become more feasible; for example, the use of deferred consent in patient recruitment and rapid high-throughput gene expression quantification should enable more representative cohorts to be recruited and profiled both within the United Kingdom and internationally.

The ideal cohort design to investigate sepsis-dependent eQTL effects would include samples taken from the same individuals prior to their septic event to allow a within-study interaction model. Given that this is not feasible, we leveraged publicly available summary statistics[44,45] to compare effect sizes[46] between sepsis and health. However, technical differences between studies, such as cohort ancestry, sample size, experimental platform, the set of variants assessed, and the availability of summary statistics, can confound this comparison. We therefore minimized the impact of these variables by matching them to our cohort as closely as possible. In addition, given the potential utility of SRS trajectories over time for prognostication and understanding treatment responses, more comprehensive serial sampling should be employed in future studies. Finally, there is currently very limited *in vivo* binding data on different TFs in primary cells, particularly in the disease context, and any single TF may have both activating and repressive effects across different genes and contexts. We have therefore used curated regulon and predicted binding sites to prioritize candidate regulatory drivers that could be suitable for longer-term functional follow-up. With this approach, we are limited to the factors included in each tool, so we may have missed an opportunity to discover additional relevant factors.

## Conclusion

In conclusion, our eQTL interaction approach has identified factors putatively linking host genetic variation, cell subtypes, and the individual transcriptomic response to infection. Understanding the regulatory networks underlying patient heterogeneity could inform the development of immunomodulatory treatments and personalized medicine in sepsis.

## STAR★METHODS

Detailed methods are provided in the online version of this paper and include the following:

## SUPPLEMENTAL INFORMATION

## CONSORTIA

The following GAinS investigators, listed alphabetically by institution, were involved in patient recruitment, sample collection, or sample processing: Jenni Addison, Helen Galley, Sally Hall, Sian Roughton, Jane Taylor, Heather Tennant, Nigel Webster, Achyut Guleri, Natalia Waddington, Dilshan Arawwawala, John Durcan, Christine Mitchell-Inwang, Alasdair Short, Susan Smolen, Karen Swan, Sarah Williams, Emily Errington, Tony Gordon, Maie Templeton, Marie McCauley, Pyda Venatesh, Geraldine Ward, Simon Baudouin, Sally Grier, Elaine Hall, Charley Higham, Jasmeet Soar, Stephen Brett, David Kitson, Juan Moreno, Laura Mountford, Robert Wilson, Peter Hall, Jackie Hewlett, Stuart McKechnie, Roser Faras-Arraya, Christopher Garrard, Paula Hutton, Julian Millo, Penny Parsons, Alex Smiths, Duncan Young, Parizade Raymode, Jasmeet Soar, Prem Andreou, Sarah Bowrey, Dawn Hales, Sandra Kazembe, Natalie Rich, Emma Roberts, Jonathan Thompson, Simon Fletcher, Georgina Glister, Melissa Rosbergen, Jeronimo Moreno Cuesta, Julian Bion, Ronald Carrera, Sarah Lees, Joanne Millar, Natalie Mitchell, Annette Nilson, Elsa Jane Perry, Sebastian Ruel, Jude Wilde, Heather Willis, Jane Atkinson, Abby Brown, Nicola Jacques, Atul Kapila, Heather Prowse, Martin Bland, Lynne Bullock, Donna Harrison, Anton Krige, Gary Mills, John Humphreys, Kelsey Armitage, Shond Laha, Jacqueline Baldwin, Angela Walsh, Nicola Doherty, Stephen Drage, Laura Ortiz-Ruiz de Gordoa, Sarah Lowes, Charley Higham, Helen Walsh, Verity Calder, Catherine Swan, Heather Payne, David Higgins, Sarah Andrews, Sarah Mappleback, Charles Hinds, D. Watson, Eleanor McLees, Alice Purdy, Martin Stotz, Adaeze Ochelli-Okpue, Stephen Bonner, Iain Whitehead, Keith Hugil, Victoria Goodridge, Louisa Cawthor, Martin Kuper, Sheik Pahary, Geoffrey Bellingan, Richard Marshall, Hugh Montgomery, Jung Hyun Ryu, Georgia Bercades, Susan Boluda, Andrew Bentley, Katie Mccalman, Fiona Jefferies, Alice Allcock, Katie Burnham, Emma Davenport, Cyndi Geoghegan, Julian Knight, Narelle Maugeri, Yuxin Mi, and Jayachandran Radhakrishnan.

## ACKNOWLEDGMENTS

We thank all the patients, patient families, nurses, and clinicians who participated in the UK Genomic Advances in Sepsis (GAinS) study. We are grateful to Giuseppe Scozzafava for maintaining the sample biobank and to the Sanger Institute's Scientific Operations team for generating the RNA-seq data. We thank Guillaume Noell and the Wellcome Sanger Institute's Human Genetics Informatics (HGI) team for mapping the bulk RNA-seq reads. We are grateful to all members of the Davenport lab and particularly to Alex Tokolyi for advice on the mediation approach. The Genotype-Tissue Expression (GTEx) Project was supported by the Common Fund of the Office of the Director of the National Institutes of Health and by NCI, NHGRI, NHLBI, NIDA, NIMH, and NINDS. Data used for specific analyses described in this article were obtained from the GTEx portal on October 23, 2019. This work was funded, in whole or in part, by the Wellcome Trust Investigator Award (204969/Z/16/Z) (J.C.K.) and Wellcome Trust core funding to the Wellcome Sanger Institute (grants 206194 and 108413/A/15/D) and the Medical Research Council

(MR/V002503/1) (J.C.K. and E.E.D.). N.M. received Master's funding from the Churchill Scholarship from the Winston Churchill Foundation. For the purpose of Open Access, the author has applied a CC BY public copyright license to any author accepted manuscript version arising from this submission.

## AUTHOR CONTRIBUTIONS

E.E.D., K.L.B., and J.C.K. conceptualized the study. E.E.D. and J.C.K. supervised the study. Funding was acquired by J.C.K., E.E.D., and N.S. C.J.H., J.C.K., S.M., and the GAinS investigators recruited the patients. K.L.B., Y.M., C.G.G., and A.J.K. performed the experimental work. A.J.K., K.C.-G., and S.M. provided resources. K.L.B., N.M., W.L., and P.Z. performed the data analysis. K.L.B., E.E.D., N.M., W.L., and J.C.K. interpreted the results. K.L.B., E.E.D., N.M., and W.L drafted the manuscript. Review and editing were carried out by all authors who read, provided input on, and approved the paper.

## DECLARATION OF INTERESTS

The authors declare no competing interests.

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

## STAR★METHODS

### KEY RESOURCES TABLE

| REAGENT or RESOURCE | SOURCE | IDENTIFIER |
|---|---|---|
| **Deposited data** | | |
| RNA-sequencing data | Cano-Gamez et al[13] | EGA: EGAD00001008730 |
| Genotyping data | Davenport et al[7], this paper | EGA: EGAD00001015369 |
| GTEx v8 whole blood summary statistics | GTEx Portal[44] | https://gtexportal.org/home/downloads/adult-gtex/overview |
| eQTLgen phase I cis-eQTL summary statistics | eQTLgen consortium website[45] | https://www.eqtlgen.org/cis-eqtls.html |
| **Software and algorithms** | | |
| Plink 1.9 | Chang et al[97] | https://www.cog-genomics.org/plink/1.9/ |
| Imputation preparation script | W. Rayner | http://www.well.ox.ac.uk/~wrayner/tools/ |
| Sanger Imputation Service | Wellcome Sanger Institute[98] | https://imputation.sanger.ac.uk/ |
| liftOver | Hinrichs et al[99] | https://genome.ucsc.edu/cgi-bin/hgLiftOver |
| King v 2.3.2 | Manichaikul et al[100] | https://www.kingrelatedness.com |
| HLA personalised mapping algorithm | Lee et al[101], this paper | https://github.com/davenportlab/HLApm |
| arcasHLA 0.2.0 | Orenbuch et al[102] | https://github.com/RabadanLab/arcasHLA |
| IMGT/HLA database v 3.42.0 | EMBL-EBI | https://www.ebi.ac.uk/ipd/imgt/hla/ |
| HIBAG v.1.4 | Zheng et al[103] | https://bioconductor.org/packages/release/bioc/html/HIBAG.html |
| GCTA v.1.93.3 | Yang et al[33] | https://yanglab.westlake.edu.cn/software/gcta/#Overview |
| R v.3.6 | The R Project for Statistical computing | https://www.r-project.org/ |
| R package: coloc | Giambartolomei et al[40] | https://cran.r-project.org/web/packages/coloc/index.html |
| R package: genpwr | Moore et al[104] | https://CRAN.R-project.org/package=genpwr |
| R package: peer | Stegle et al, Parts et al[105,106] | https://github.com/PMBio/peer |
| R package: lme4 | Bates et al[107] | https://cran.r-project.org/web/packages/lme4/index.html |
| eigenMT | Davis et al[108] | https://github.com/joed3/eigenMT |
| R package: interactions | Long[109] | https://cran.r-project.org/web/packages/interactions/index.html |
| R package: mashr | Urbut et al[46] | https://cran.r-project.org/web/packages/mashr/index.html |
| R package: XGR | Fang et al[110] | https://github.com/hfang-bristol/XGR |
| SNP2TFBS | Kumar et al[54] | https://epd.expasy.org/snp2tfbs/ |
| R package: JASPAR2014 | Mathelier et al[55] | https://bioconductor.org/packages/release/data/experiment/html/JASPAR2014.html |
| R package: JASPAR2024 | Rauluseviciute et al[111] | https://bioconductor.org/packages/release/data/annotation/html/JASPAR2024.html |
| R package: decoupleR | Badia-I-Mompel et al[57] | https://bioconductor.org/packages/release/bioc/html/decoupleR.html |
| R package: WGCNA | Langfelder et al[59] | https://cran.r-project.org/web/packages/WGCNA/index.html |
| R package: spqn | Wang et al[112] | https://bioconductor.org/packages/release/bioc/html/spqn.html |
| R package: xCell | Aran et al[60] | https://github.com/dviraran/xCell |

*(Continued on next page)*

*Continued*

| REAGENT or RESOURCE | SOURCE | IDENTIFIER |
|---|---|---|
| R package: survival | | https://cran.r-project.org/web/packages/survival/index.html |
| R package: mediation | Tingley et al[61] | https://cran.r-project.org/web/packages/mediation/index.html |
| QTL mapping code | This paper | Zenodo: https://doi.org/10.5281/zenodo.10987496 |
| Other | | |
| Open Targets Genetics portal | | https://genetics.opentargets.org/ |

## RESOURCE AVAILABILITY

### Lead contact
Further information and requests for resources should be directed to and will be fulfilled by the lead contact, Emma Davenport (ed5@sanger.ac.uk).

### Materials availability
This study did not generate new unique reagents.

### Data and code availability
- Raw de-identified RNA-sequencing and processed genotyping data derived from human samples have been deposited at the European Genome-phenome Archive (EGA), and accession numbers are listed in the Key resources table. They are available upon request via data access committee if access is granted. In addition, processed gene expression data are available as Supplementary Tables.
- All original code has been deposited at Zenodo and is publicly available as of the date of publication. DOIs are listed in the Key resources table.
- Any additional information required to reanalyze the data reported in this paper is available from the lead contact upon request.

## EXPERIMENTAL MODEL AND STUDY PARTICIPANT DETAILS

### Human study participants
This observational study comprised adult sepsis recruited in the UK, with available demographic data and individual level covariates summarised in supplementary tables. Ethics approval was granted nationally and locally for individual participating centres, and we obtained informed consent from the patient or their legal representative.

## METHOD DETAILS

### Data generation and processing
#### Cohort description
We recruited adult patients (>18 years old) to the UK Genomic Advances in Sepsis (GAinS) study (NCT00121196) from 34 intensive care units (ICUs) between 16/11/2005 and 30/05/2018[7]. Ethics approval was granted nationally and locally for individual participating centres, and we obtained informed consent from the patient or their legal representative. Inclusion criteria were sepsis diagnosed according to ACCP/SCCM guidelines due to community acquired pneumonia (CAP) or faecal peritonitis (FP). CAP was defined as febrile illness associated with cough, sputum production, breathlessness, leukocytosis and radiological features of pneumonia acquired in the community or within two days of admission to hospital. FP was defined as inflammation of the peritoneal membrane secondary to faecal contamination, diagnosed by laparotomy. Exclusion criteria were immunosuppression, admission for palliative care only, and pregnancy.

#### Sample collection and processing
We extracted DNA from buffy coat or whole blood samples using the Qiagen DNA extraction protocol, the automated Maxwell Blood purification kit (Promega), or the QIAamp Blood Midi kit protocol (Qiagen). We took serial samples for RNA extraction on the first, third, and/or fifth day after ICU admission. We isolated the total leukocyte population at the bedside using the LeukoLOCK filter system (Life Technologies), and extracted purified RNA using the Total RNA Isolation Protocol.

## QUANTIFICATION AND STATISTICAL ANALYSIS

### Genotyping data generation and processing

We had previously generated genotyping data for 295 CAP patients and 730,525 SNPs using the Illumina HumanOmniExpress BeadChip[7]. We genotyped a further 655 patients at the Wellcome Sanger Institute using the Infinium CoreExome BeadChip (551, 839 SNPs), and 307 patients (including 38 repeat of samples that failed QC previously) at the Wellcome Centre for Human Genetics using the Infinium Global Screening Array BeadChip (654,027 SNPs).

We performed genotyping QC within each batch in Plink 1.9[97] according to the methods described in Anderson et al 2010[113]. We excluded samples on the basis of discordant sex information, proportion of missing genotypes > 0.02, outlying heterozygosity rate, identity by descent (Pi_hat >=0.1875), and detection of sample mix-ups through comparison to RNAseq on the same patient (details below)[114]. We excluded variants if they had a missing data proportion > 0.05, MAF<0.01, and Hardy-Weinburg equilibrium $P<1\times10^{-5}$.

### Imputation

We imputed each of the three genotyping data sets using the Haplotype Reference Consortium (HRC) release 1.1 panel and the Sanger Imputation Service[98], following checks for strand, alleles, position, ref/alt assignments, and frequency differences versus the HRC (http://www.well.ox.ac.uk/~wrayner/tools/). Genotypes were phased using Eagle2 (v2.0.5[115]) and imputed using Positional Burrows-Wheeler Transform (PBWT[116]). We removed SNPs with an imputation info score <0.9, combined the data sets, and converted positions to build 38 coordinates using liftOver[99]. Finally, we excluded individuals with missingness >2% and SNPs with missingness ≥2%. This resulted in a final data set of 12,412,067 SNPs and 1,168 individuals. Projecting the samples onto a PCA of 1KGP using King[100], we found that the vast majority of data points clustered together with European ancestry individuals (Figure S2A). We did not remove any individuals based on this PCA, but calculated genotyping PCs on the combined GAinS genotyping data set using Plink[97] and SNPs with MAF >1% and included genetic PCs in downstream analyses.

### RNA-sequencing and data processing

As described in Cano-Gamez et al (2022)[13], we prepared stranded cDNA libraries for 909 samples from 695 sepsis patients using NEB Ultra II Library Prep kits (Illumina) with NEBNext Poly(A) mRNA Magnetic isolation, and sequenced them across 3 S4 flow cells using an Illumina Novaseq 6000 system. We obtained a median of 40M 100bp paired-end reads per sample. We aligned reads to the reference genome (GRCh38.99) using STAR v2.7.3a[117] (ENCODE recommended parameters) and quantified gene expression using featureCounts v2.0.0[118] (https://github.com/wtsi-hgi/nextflow-pipelines/tree/2e5ac3cee33ca2a1ced2943bb7e366a7771a4d3c).

### Construction of personalised HLA references

To improve accuracy of HLA gene expression quantification, we created a personalised HLA reference for each patient (https://github.com/davenportlab/HLApm). We imputed two-field resolution HLA types with arcasHLA 0.2.0[102] using the IMGT/HLA database v 3.42.0 for fourteen HLA genes (*HLA-A, -B, -C, -E, -F, -DMA, -DMB, -DPA1, -DPB1, -DQA1, -DQB1, -DRA, -DRB1, and -DRB3*) and HIBAG v.1.4[103] using a pre-trained multi-ethnic model for seven HLA genes (*HLA-A, -B, -C, -DPB1, -DQA1, -DQB1, and -DRB1*), from RNA-seq and genotyping data respectively. We then assessed the concordance between the RNA-seq and genotyping-based predictions as follows; 0: discordant calling for both alleles of a gene, 1: concordant call for one allele of a gene, 2: concordant calling for both alleles of a gene. We found greater concordance for HLA Class I alleles than class II (95.4% vs 91.2%) with >=1 concordant calls. We used HLA alleles from arcasHLA in cases with discordant calling. For alleles with missing calls from arcasHLA due to low RNA-seq read depth, alleles predicted by HIBAG were used. These imputed alleles were then used to define personalised reference sequences for HLA gene expression re-quantification.

In the IMGT/HLA database, untranslated exons are often missing, and UTR annotation is either absent or variable across alleles. As gene expression would be underestimated if reads originating from these regions were not mapped, we completed the personalised gene references by extending both the 5' and 3' sequences of each allele from IMGT based on primary reference sequences. In addition, for HLA-DRB2/4/7/8, we included two alternative reference sequences (GL000255 and GL000256) as these genes are missing in the primary reference sequences. Finally, we adjusted gene coordinates to reflect these extended allele sequences.

### HLA re-mapping with personalised references

For each sample, we extracted reads mapping to the MHC (chr6:28500000-33400000, GRC38) and unmapped reads from the initial genome-wide mapping (see above). We aligned the extracted reads to the personalised reference sequences using STAR v2.7.3a. Depending on the heterozygosity of each gene, this produced up to 28 aligned BAM files per sample. We proceeded to assign reads to genes, following the method used in AltHapAlignR[101]. We merged all alignments from a given sample, retaining only uniquely mapped reads from each mapping run (bam file). We required that both reads in a pair were aligned to the exon region with at least one base on the same allele. For reads aligned to multiple genes, we compared the editing distance (NM value) between the reads and reference sequences and assigned the reads to the gene with the lowest editing distance. We discarded reads with the same editing distance in different genes. As the reads inputted to this HLA re-mapping included reads initially mapped to any region in the MHC and not just to the 14 HLA genes with personalised references, we also used this approach to assign reads previously mapped to any

other MHC gene and remapped here. For reads with multiple gene assignments, we reassigned each read to the gene with the lowest editing distance. Finally, gene counts were recalculated for all MHC region genes.

### Gene expression QC

We used MBV from QTLTools[114] to identify mismatches between the genotyping and RNA-seq gene expression data. A mismatch, with or without sex mismatch in the genotyping or gene expression compared to the clinical information, could indicate a sample mixup. If we could determine the true sample identity, these were resolved; otherwise we excluded the affected patients from the analysis.

We excluded 45 samples following QC including mapping rate, PCA outliers, resolution of sample mix-ups and detection of contamination using VerifyBamID[119]. This resulted in a data set of 864 samples from 667 patients, including 135 samples repeated from previously published microarray data[7]. After targeted remapping of MHC region genes (see above), we filtered out features that did not have at least 10 reads in at least 5% of samples, retaining 20,412 genes for downstream analysis. We then normalised the data, using the trimmed mean of M-values method[120] to estimate size factors for calculating counts per million, and added a pseudocount of 1 for log-transformation. The normalised gene expression data are available in Table S25.

### GWAS
#### SRS assignment

All patient gene expression samples had Sepsis Response Signature (SRS) assignments as described in Cano-Gamez et al[13]. Briefly, the SepstratifieR package includes a reference dataset with SRS assignments from the original clustering analysis[7,12] and uses the expression of 7 pre-selected marker genes to assign SRS membership to new samples. New gene expression data are aligned to the GAinS reference set using the k-nearest neighbours algorithm, then random forest models are employed to predict SRS and SRSq. We filtered our genotyping cohort to 997 patients with SRS assignments on at least 1 time point from any of microarray, RNA-seq, and qPCR. Where SRS1 status was available from multiple assays on the same time point, we found discrepant assignments for 10 samples. In these cases, we used SRS1 status from RNA-seq preferentially followed by microarray, resulting in 516 SRS1 samples and 818 non-SRS1 samples. We categorised patients as "Ever SRS1" if one or more time points (n1=736 patients, n2=185, n3=76) were assigned to SRS1. Our final cohort comprised 440 ever SRS1 patients and 557 never SRS1.

#### Heritability estimation

We used the GREML approach from GCTA[33,121,122] to estimate the proportion of variation in the "ever SRS1" phenotype explained by common variants (MAF>1%). We used LD-pruned imputed autosomal genotype data (Plink[97]; –indep-pairwise 50 5 0.2) to calculate the GRM, and included age, $age^2$, sex, and 7 genotyping principal components (PCs) as covariates in the model as fixed effects.

#### Genome-wide association study

We performed a GWAS for SRS1 ever vs never using Plink[97,123], including age, $age^2$, sex, and 7 genotyping principal components (PCs) as covariates in a logistic regression model. Independent loci were defined by clumping ($r^2<0.5$). The coloc R package[40] was used to perform colocalisation between GWAS and independent eQTL signals using default priors. The genpwr[104] R package was used to estimate sample sizes required for replication with >80% power, based on the odds ratios and minor allele frequencies of the lead SNPs from the GWAS loci and the case rate observed in this cohort.

### eQTLs
#### cis-eQTL mapping

We restricted the genotyping data set to the 638 patients with RNA-seq gene expression data available, and to biallelic variants with MAF ≥ 1% in this subset of individuals. Genotypes were coded as 0, 1, or 2 according to the number of copies of the minor allele carried by each patient. We filtered the RNA-seq data set to the 823 samples from 638 patients with genotyping data available. We calculated 30 PEER (probabilistic estimation of expression residuals) factors[105,106] on the reduced gene expression dataset used for eQTL mapping, holding out rank-based inverse normal transformed (INT) clinically measured cell proportions (neutrophils, monocytes, lymphocytes), SRS1 status, site of infection (CAP/FP), and 7 genotyping PCs. For 5 samples with missing cell proportion information, we used the median value for the cohort (Table S26).

We tested each autosomal gene for associations with SNPs within 1Mb of their transcriptional start site using the lme4 R package[107]. We used a linear mixed model to test each SNP-gene pair for an additive effect of genotype, including a random intercept effect for individual so that up to three serial samples per patient might be included to increase power[28]. We included the first seven principal components (PCs) of the genotyping data and the first 20 PEER factors of the gene expression data in the model to correct for systematic effects. This was based on including increasing numbers of PEER factors in the model to determine the "elbow" where adding additional factors did not result in a large increase in detected eQTLs (Figure S3). We also included the covariates held out from the PEER factor calculation in the model.

We identified genes with a significant eQTL using a hierarchical approach for multiple testing correction[124]. Nominal p-values for the effect of the variant were obtained from a likelihood ratio test, using the anova function implemented in the lme4 package[107]. We used eigenMT[108] to estimate the number of independent tests performed per gene, and from this calculated locally adjusted p-values for the peak association for each gene by Bonferroni correction. We then performed global p-value adjustment across the peak associations for all genes with the Benjamini-Hochberg method, and used a false discovery rate (FDR) threshold of 0.05 to identify

**Cell Genomics**
*Article*

genes with at least one significant eQTL (eGenes). We defined additional significant SNP-gene pairs for these eGenes as those with a nominal p-value less than the per-gene p-value threshold corresponding to the global FDR threshold of 0.05.

### Conditional analysis

We then defined conditionally independent associations for each significant eGene using forward stepwise regression and backwards selection[37]. First, we repeated the *cis*-eQTL mapping, adjusting for the most significant SNP identified in the first round and using the gene-level significance threshold to identify independent secondary *cis*-eQTL signals. We repeated this iteratively, accounting for all the lead variants discovered in previous iterations to learn the number of independent signals. Once no new variants were identified, we terminated the forward stage and performed a backwards selection step, in which we re-tested each signal discovered in the forward step in turn. We included all other forward pass variants as covariates in the eQTL scan, and retained the most significant variant as the lead association for each independent signal. If we did not detect any significantly associated variants for a given signal in the backwards step, then the signal was dropped. Finally, we calculated effect sizes for each independent signal by fitting a model including all significant signal SNPs.

### Overlap with external GWAS results

We queried the Open Target Genetics portal[34,35] to look up previously reported associations for immune/inflammatory diseases and blood cell traits for SNPs of interest. We also obtained a list of SNPs associated with mortality in sepsis from Le et al 2019[125] and Hernandez-Beeftink et al 2022[26] to compare to our SRS GWAS results.

### Interaction analysis

We tested the signal variants from the conditional eQTL analysis for interactions with SRS1 status, diagnosis (CAP vs FP) and cell proportion (monocyte, lymphocyte, or neutrophil), whilst also correcting for the effect of any independent signal variants. For comparison, we similarly tested for eQTL interactions with sex. As this variable was not included as a covariate in the original eQTL mapping, we first confirmed that the autosomal SNP-gene pair remained significant with sex added to the model, and then tested for an interaction effect.

SNPs were only tested if there were at least two minor allele homozygote individuals in each subgroup or each half of all cell proportion distributions being tested. Interaction p values were calculated using a likelihood ratio test and corrected for multiple testing using the Benjamini-Hochberg method, and significant interactions identified using an FDR threshold of 0.05. To assess whether we had detected more interaction effects than would be expected by chance, we permuted SRS status across all samples or diagnosis across patients and repeated the interaction analysis 1000 times. Plots were made using the interactions R package[109].

### Definition of sepsis-dependent eQTLs

We accessed the GTEx v8 whole blood summary statistics for European individuals through the GTEx Portal[44], and the eQTLgen cis eQTL summary statistics from the eQTLgen Consortium website[45]. We compared z scores and significance for the lead SNPs for eGenes identified in GAinS, where the same SNP-gene pair was tested in the public dataset and the assessed allele could be disambiguated from strand designation. Significance in GTEx was defined as the gene having a qvalue < 0.05 in the *cis*-eGenes output, and the nominal p-value for the SNP being below the nominal p-value threshold for that gene.

We also used mashr[46] to compare beta values between GAinS eGenes and GTEx, following the approach outlined in the mashr eQTL analysis vignette. Mashr (Multivariate Adaptive Shrinkage in R) compares effects on the same SNP-gene pairs across multiple conditions, employing empirical Bayes methods to estimate patterns of similarity. We used GTEx[44] for this analysis, due to the greater technical similarity to GAinS and availability of beta values and standard errors. Briefly, we used a random 5% subset of the SNP-gene pairs tested in both GAinS and GTEx ("random" tests) to learn the correlation structure across null tests and the lead SNPs for each significant eGene identified in sepsis ("strong" tests) to learn data-driven covariance matrices. We fitted the mashr model to the "random" tests, then used it to calculate posterior summaries on the "strong" tests from which to identify differing effects across conditions. We categorised the eQTLs significant in sepsis (mashr lfsr<0.05, n=8,122) as "shared" if the mashr posterior effect size was in the same direction as and within a factor of 0.5 of the GTEx effect size, or "context-dependent" if this was not the case. We classed those significant in both GAinS and GTex but with opposite directions of effects as "opposite effect" (n=53), and divided the remainder into those with bigger effects and/or only significant in sepsis ("sepsis-magnified", n=854, 10%) or in HV ("sepsis-dampened", n=1,272)

### Enrichment and network analysis

We used the XGR R package[110] and Reactome annotations for pathway analysis, using Fisher's Exact Test with the background defined as all genes tested in a given analysis. For example, when testing eGenes with an SRS interaction for enriched pathways, the background was all eGenes tested for an SRS interaction effect. P-values were adjusted using the Benjamini-Hochberg FDR procedure and an FDR threshold of 0.05 used to determine significance. We used a comparable approach to test for enrichment of gene sets defined in our results (e.g. sepsis-magnified eGenes vs different types of interactions). We also used XGR to test for enrichment of interaction eSNPs vs non-interaction eSNPs in Roadmap Epigenomics Core 15-state Genome Segmentation annotations[47] for primary peripheral immune cells with a one-tailed binomial test, and for exploration of subnetworks within eGenes with interactions using default parameters.

### Transcription factor enrichment analysis

We used the SNP2TFBS[54,55] database to identify instances of eSNPs significant in sepsis introducing or interrupting predicted transcription factor binding sites. As the lead eSNP is not necessarily the causal variant, we first expanded the query SNPs to all SNPs in LD ($r^2 \geq 0.8$) with the signal SNPs in our cohort. We restricted the TF motifs considered to those annotated by Jaspar 2014

(JASPAR2014 R package[126]) as being defined in Homo sapiens. Given SNP2TFBS used an older version of the JASPAR motif set, we checked alignment of motifs with the JASPAR 2024 database[111] and removed 3 motifs that were not present in the latter (n=124/205 motifs included in SNP2TFBS retained). We then collapsed the results for each independent eQTL signal, scoring each motif as having $\geq 1$ or 0 binding sites altered by the signal SNP or its LD proxies. We then tested for enrichment of each TF motif with at least one SNP overlap, amongst eQTLs with a significant interaction effect compared to eQTLs with no significant interaction using a one-sided Fisher's exact test. We adjusted the p-values obtained across all motifs using the Benjamini-Hochberg method. We permuted eGene interaction status 1000 times and repeated the enrichment analysis to determine how many enriched motifs we would expect to see by chance.

We inferred transcription factor activity from our RNA-seq dataset using the DoRothEA regulons[56], restricted to human TFs with evidence levels A-C and a minimum regulon size of 5. We calculated a consensus activity score as implemented in the decoupleR R package[57]. We tested for differential activity between SRS groups using a linear mixed model with a random intercept for individual, adjusting the p-values for multiple testing using the Benjamini-Hochberg method. We permuted SRS status and repeated the differential activity analysis to determine how many TFs we would expect to have differential activity by chance.

We matched the transcription factors for which we had inferred activity to the motifs assessed by SNP2TFBS[54] to pinpoint TFs with evidence of relevance to SRS in both analyses. Using the first available sample for each patient, we used Spearman correlation to identify relationships between TF activity and cell proportions estimated using CIBERSORTx[127] based on a sepsis single cell reference set[14].

## Co-expression modules
### Co-expression module discovery
We identified co-expression modules in the gene expression data using the WGCNA R package[58] as follows. First, to control for technical variation during module identification[128], we regressed the top 20 gene expression PCs out from the logCPM gene expression matrix. So that we only considered between-individual correlation, we replaced the gene expression value in each sample from the same individual with their mean gene expression[129]. We then calculated the biweight midcorrelation matrix for the residual gene expression to generate a similarity matrix using the bicor function from the WGCNA R package[58]. We used spatial quantile normalisation implemented in the spqn R package[112] to account for the mean-correlation bias in the similarity matrix, and applied the normalize_correlation function to the similarity matrix with 21 blocks of size 1000 and block 18 as the reference group.

We determined a soft threshold value of 4 for the similarity matrix using the pickSoftThreshold function. We used this soft threshold to build an unsigned adjacency matrix using the adjacency function, which we used in turn to calculate the topological overlap metric (TOM) matrix using the TOMsimilarity function. We applied the dynamic tree cut algorithm included in the WGCNA package as the cutreeDynamic function to generate modules with default parameters and a minimum cluster size of 10. Similar modules were merged based on the similarity of their module eigengenes using the mergeCloseModules function with a cut height of 0.1. For a module, the eigengene was defined as the first PC of the gene expression data of the genes present in the module. We calculated the module eigengenes for the final set of modules using the moduleEigengenes function.

### Module annotation
We tested for enrichment of Reactome pathways among module member genes using XGR, and for downstream targets of different transcription factors using DoRothEA regulons[56] and Fisher's Exact Test. In each case, p-values were adjusted using Benjamini-Hochberg FDR correction and a threshold of 0.05 used. The set of expressed genes was used as the background for enrichment.

We used cell markers from the xCell R package[60] and a single-cell RNA-seq sepsis dataset[14] to identify cell-type-specific modules. The xCell signatures were derived based on differential gene expression from large transcriptomic studies of individual cell types and built to minimise classification error. Enrichment of gene signatures was performed using a hypergeometric test using the phyper function in R. The entire set of expressed genes was considered the background for enrichment. P-values were corrected using the Benjamini-Hochberg FDR procedure. Since multiple transcriptomic studies assayed the same cell types in xCell, one cell type often had multiple signatures. The median odds of enrichment per cell type in xCell were reported for any signatures that passed a q-value cutoff of 0.05. In contrast, each cell type in the Kwok et al. 2023 study had one set of markers. Signatures for cell types passing a q-value cutoff of 0.05 were reported. Enrichment was calculated as the ratio between the proportion of signature genes in the module and the proportion of signature genes in the entire set of expressed genes.

We tested for association between 28-day survival and each eigengene using a Cox proportional hazards model, as implemented in the survival R package using the coxph function. For each patient, the value of the eigengene at the last time point assayed was used as a predictor for the survival function. For all other patient phenotypes (cell proportions, SRS1 status, diagnosis and time point), we used a linear mixed model to test for differences in eigengene expression between groups. Measured cell proportions were inverse normal transformed and missing values replaced with the median value. P-values were corrected for multiple testing using the Benjamini-Hochberg procedure.

### Module QTL
We tested the 12,335 unique signal eSNPs from our conditional eQTLs with >3 minor allele homozygotes for associations with all module eigengenes. As in the cis-eQTL analysis, we included seven genotyping PCs, 20 PEER factors, SRS status (SRS1 versus non-SRS1), diagnosis (CAP versus FP), and transformed cell proportions as fixed-effect covariates. A genome-wide threshold of

**Cell Genomics**
**Article**

$3.82 \times 10^{-8}$ was used based on a Bonferroni correction accounting for the number of SNPs and number of modules tested. We defined loci for each module by constructing 1 Mb windows around each module QTL SNP and merging those that overlapped.

### ModQTL replication

We sought to replicate the 32 lead modQTL associations using our previously published microarray gene expression data (Tables S27 and S28)[13]. First, for each module gene set with at least 5 genes passing QC in the microarray (Modules 97, 88, 103 were not replicable), we computed eigengene values in the full microarray cohort (n=676 samples) using the svd R function. We assessed eigengene correlation across technologies based on 135 patient samples included in both the microarray and the RNA-seq cohort (Spearman's rho). We confirmed that all lead modQTL SNPs had a MAF $\geq 1\%$ in the non-overlapping microarray sample set with genotyping data (n=506 samples from 361 patients). We then tested for replication of the association of the module eigengenes with the lead modQTL SNP in these samples using the same model design as above (p-value threshold 0.05). We used the signs of the beta values and of the correlation coefficients from the 135 repeated samples to assess concordance of direction of effects between the RNA-seq and microarray cohorts.

### ModQTL sensitivity analysis

We recalculated the module eigengenes in the RNA-seq cohort excluding all conditional *cis*-eGenes for which any associated SNPs were lead eSNPs, and retested for association with the lead module SNP using the same model as above. Significance was determined using the same genome-wide threshold.

### Mediation analysis

We then tested for mediation of associations between the lead modQTL eSNP (treatment) and the recalculated eigengene (outcome) by the modQTL SNPs' target *cis*-eGene(s) (mediator, each tested separately) using the mediation R package[61]. All covariates used in the main modQTL model were included. A quasi-Bayesian approximation was used for confidence intervals with 1000 simulations, and effect sizes represent the effect of 1 additional copy of the minor allele.

