## [Document S2. Transparent peer review records for Burnham et al. · Cell Genomics]

eQTLs identify regulatory networks and drivers of variation in the individual response to sepsis

Katie L. Burnham, Nikhil Milind, Wanseon Lee, Andrew J. Kwok, Kiki Cano-Gamez, Yuxin Mi, Cyndi G. Geoghegan, Ping Zhang, GAINs Investigators, Stuart McKechnie, Nicole Soranzo, Charles J. Hinds, Julian C. Knight, Emma E. Davenport

Summary

Initial submission: Received : Nov 07, 2023

Scientific editor: Sara Rohban

First round of review: Number of reviewers: 3
Revision invited : Jan 02, 2024
Revision received : Mar 27, 2024

Second round of review: Number of reviewers: 3
Accepted : May 28, 2024

Data freely available: YES

Code freely available: YES

This transparent peer review record is not systematically proofread, type-set, or edited. Special characters, formatting, and equations may fail to render properly. Standard procedural text within the editor's letters has been deleted for the sake of brevity, but all official correspondence specific to the manuscript has been preserved.

Referees' reports, first round of review

Reviewer #1:

The authors present an extensive set of analyses in a well written manuscript that aims to tease out the heterogeneity of sepsis. This is a critical approach as the diversity of sepsis presentations, courses, and responsiveness to intervention is a massive hurdle in the development effective treatments. Specifically, these analyses characterize the genetic variation and transcription features that delineate sepsis subtypes. Overall, we need more authors to do this type of detailed/nuanced work in phenotyping and context-dependent omics. Bigger sample sizes alone will not suffice, we need better characterized and more homogenous outcome subtypes to analyze. A few points deserve consideration:

1) It would help some readers to see a table 1 style presentation of the basic study group characteristics (sex, age, CAPvsFP, etc.). This would be helpful even if it is only available in the supplemental materials.

2) The NR2F6 findings may deserve a bit more attention in the discussion. In a recent GWAS of Neonatal Sepsis in Europeans, an intronic SNP in NR2F6 was a top hit, but only after restricting the analysis to a subset of sepsis cases (bacterial culture confirmed sepsis -<https://pubmed.ncbi.nlm.nih.gov/35835848/> - see Table S4 in the supplemental materials). This overlap with prior findings deserves mention and perhaps the authors could add a sentence of two about the infectious agents in their work. Are there ways to identify the "causative bug" or "category of causative bug" in their participants. Could this be somehow gleaned (even crudely) with the CAP and FP distinctions? The authors do not overreach in their speculations (and this is good) but some of the available NR2F6 literature suggests it may be more important certain sepsis subtypes (e.g. bacterial peritonitis) <https://pubmed.ncbi.nlm.nih.gov/28779026/> <https://pubmed.ncbi.nlm.nih.gov/36052079/> . . . this may be a direction for future inquiry.

3) While not necessarily required for this manuscript, two additional contexts may deserve mention as future directions. The first is the relative timing of SRS1 in these patients. There are many ways to parse this out but the subsections of "ever SRS1" might be important (do temporal trajectories of SRS1 development in relation to other clinical features predict outcomes). The second piece of context is biological sex. The genetics, physiology, and incidence of sepsis are known to differ between the sexes. <https://pubmed.ncbi.nlm.nih.gov/24966193/> <https://pubmed.ncbi.nlm.nih.gov/35835848/> <https://pubmed.ncbi.nlm.nih.gov/28963611/> <https://pubmed.ncbi.nlm.nih.gov/19996327/> <https://pubmed.ncbi.nlm.nih.gov/14223226/> In short, studies of sepsis should be stratified by sex when possible. Since so much work has already been presented here, the authors could avoid doing this, by simply noting that this should be done in future work.

4) The excel file in the supplement could benefit from better labels (e.g. in table 2 - what is S05, S001, etc.)

Reviewer #2:

eQTLs identify regulatory networks and drivers of variation in the individual response to sepsis

In this manuscript, Burnham and colleagues discuss eQTL mapping in a cohort of sepsis patients. The paper is very well written and statistically rigorous, leading to exciting and trustworthy insights into the disease. I have a couple of comments that could further improve the work, but overall I am very happy with the status already.

Main comments

One of the main points I would like to see addressed is the classification of SRS states. I understand this method has been published before, but the current manuscript would benefit from a brief description of the methodology, what data is required to classify a person, and the characteristics of SRS1 and SRS2 (both mentioned in the text, in the context of a 'spectrum'). Lastly, it would be good to explain how the SRSq score relates to these sub types.

Second, the authors show a pretty good replication rates with a previous, smaller cohort. However, the other cohort shares almost half of its samples with the current discovery cohort. In order to get a more accurate estimate of replication, the authors should repeat this analysis without the sample overlap, by removing the shared samples from discovery or replication.

Last, regarding the Transcription factor enrichment analysis: "We restricted the TF motifs considered to those annotated by Jaspar 2014 (JASPAR2014 R package112) as being defined in Homo sapiens (n=127/205 included in SNP2TFBS)." Is it correct that you then only used 127 TF motifs for this analysis? Jaspar has had many updates since 2014, including recently to 2024. I would like to request that you use a more up to date motif set, since the database changes quite a lot over time. Alternatively, you could check for the TF motifs that you've included, how well they align with the more recent versions of the database. The reason I'm asking, is because it is important to have current motif information if you want the SNP to TFBS analysis to work effectively.

Minor comments

- While the authors explain that the eQTLs are identified in patients, but most of them can be found in blood from healthy individuals as well, they do describe the eQTLs they found as 'sepsis eQTLs' initially. I think it's fairer to use something like 'eQTLs significant in sepsis patients'.
- Population(s) in cohort is predominantly European: did you observe any outliers? How did you deal with them?
- Sometimes use magnifying, sometimes enhanced. I think would be clearer to use same terminology
- How do you interpret the enrichment of PA in SRS interaction eQTLs?
- What is FET p-value (p6)?
- Fig 1f: some point in lower right quadrant, that should be opposite direction of effect
- Fig 2b: add lines per genotype
- It would be beneficial to give a bit more info on OCEL1
- Where do the 288 TFs come from (p7)
- "suggesting they capture gene sets associated with specific biological processes" \diamond as you've shown

that these gene sets are highly correlated to cell counts, I would say this suggests the gene sets are associated with specific cell types, not with processes

- Do you have a sense of what's happening with module 71? Not enriched for cell markers, only very lightly associated with neutrophils and lymphocytes, but not SRS at all.
- Fig 4b: what does 'Outcome' refer to? Is this the survival analysis?
- Fig 4f: there are no genes highlighted in orange, I know you've highlighted the location, but it would be good to include the names SENP7 and IMPG2, and indicate that they are not part of the module
- Something weird happened with the font in methods section 'Enrichment and network analysis'.
- "We assessed eigengene correlation across technologies based on 135 patient samples included in both the microarray and the RNA-seq cohort (Spearman's rho)." \diamond I'm happy that you did this test but I missed it in the result section. What was the correlation?
- modQTL sensitivity analysis: did you remove the cis genes from the conditional lead eSNPs as well?

Reviewer #3:

This manuscript reflects a monumental body of work integrating multiple genomic datasets in a well characterized population with sepsis. It addresses numerous important questions in sepsis, building upon the current state of knowledge to ask whether DNA variation influences SRSq, whether this variation is modified by the proportion of neutrophils vs lymphocytes, whether GWAS suggestive SNPs are also eQTLs and whether these seem to be sepsis-dependent, and whether the eQTL seems to vary with the proportion of neutrophils vs lymphocytes. It further generates data about the transcription factors dysregulated in SRS1 / SRSq.

In general it is very well written, though dense and assuming a high degree of familiarity with genomic investigations. The portion describing modQTL was a little more difficult to follow; I understood it better after reading the discussion, and I suggest minor editors to more clearly explain this in the results.

The main limitation, which I think could be more clearly discussed, is that due to the availability of the multiple forms of analysis (GWAS + RNAseq), it works exclusively in the population in which SRS and SRSq were already identified, and thus there was no attempt to replicate findings in a new population. It may be that with COVID or MARS, there would be opportunities to test whether key modQTL or GWAS-eQTL variants were observed in other populations. The GAINS population is a somewhat unique sepsis population with lower mortality than many published sepsis genomic cohorts, and it would be appropriate to acknowledge that some features may be unique to this population. However, the authors did appropriately acknowledge the lack of ancestral diversity in GAINS and describe the need for future populations to more thoroughly investigate ancestral-specific findings.

Authors' response to the first round of review

We are grateful to the three reviewers for their constructive and thoughtful comments, and for their appreciation of the challenges involved in disentangling heterogeneity in sepsis.

Reviewer #1:

The authors present an extensive set of analyses in a well written manuscript that aims to tease out the heterogeneity of sepsis. This is a critical approach as the diversity of sepsis presentations, courses, and responsiveness to intervention is a massive hurdle in the development of effective treatments. Specifically, these analyses characterize the genetic variation and transcription features that delineate sepsis subtypes. Overall, we need more authors to do this type of detailed/nuanced work in phenotyping and context-dependent omics. Bigger sample sizes alone will not suffice, we need better characterized and more homogenous outcome subtypes to analyze.

A few points deserve consideration:

1) It would help some readers to see a table 1 style presentation of the basic study group characteristics (sex, age, CAPvsFP, etc.). This would be helpful even if it is only available in the supplemental materials.

1.1 We agree with the reviewer that this would be a helpful way of presenting the cohort. We have added a table as suggested, summarising the GWAS cohort as well as just the patients included in the RNA-seq eQTL dataset to the Supplementary Tables (subsequent table numbering has therefore changed throughout). We have also added individual level information used as covariates in the eQTL mapping (diagnosis, SRS status, normalised cell proportions) as final supplementary tables.

	Full GWAS cohort (n=997)	SRS1ever (n=440)	SRS1never (n=557)
SRS1ever: n (%)	440 (44%)		
Male sex: n (%)	544 (55%)	211 (48%)	333 (60%)
Age: median (range)	65 (18-92)	65 (18-89)	65 (21-92)
Source of Sepsis: n CAP (%)	657 (66%)	229 (52%)	428 (77%)
Mortality (28 days): n (%)	159 (16%)	86 (20%)	73 (13%)
EUR (self-reported): n (%)	962 (96%)	429 (98%)	533 (96%)
	RNA-seq eQTL cohort (n=638)		
Male sex: n (%)	285 (45%)		
Age: median (range)	63 (18-90)		
Source of Sepsis: n CAP (%)	420 (66%)		
Mortality (28 days): n (%)	70 (11%)		
EUR (self-reported): n (%)	611 (96%)		
	Sample level (n=864)		
	D1 (n=278)	D3 (n=301)	D5 (n=244)
SRS1: n (%)	156 (56%)	114 (38%)	59 (24%)
Neutrophil proportion: median	0.87 (0.29-0.98)	0.86 (0.18-1.00)	0.83 (0.40-1.00)
Lymphocyte proportion	0.070 (0-0.46)	0.078 (0-0.35)	0.092 (0-0.49)
Monocyte proportion	0.050 (0-0.43)	0.050 (0-0.66)	0.064 (0-0.053)

New Supplementary Table 1: Summary of key demographic and clinical characteristics of the cohorts used in a) GWAS and b) eQTL analyses.

Results:

“We have assigned SRS group membership to 997 GAinS sepsis patients with both genotyping and blood gene expression data (from qPCR, microarray or RNA-sequencing^{7,12,13}) collected at multiple time points during the first five days of ICU admission (Figure 1a, Supplementary Table 1).”

2) The NR2F6 findings may deserve a bit more attention in the discussion. In a recent GWAS of Neonatal Sepsis in Europeans, an intronic SNP in NR2F6 was a top hit, but only after restricting the analysis to a subset of sepsis cases (bacterial culture confirmed sepsis - <https://pubmed.ncbi.nlm.nih.gov/35835848/> - see Table S4 in the supplemental materials). This overlap with prior findings deserves mention and perhaps the authors could add a sentence of two about the infectious agents in their work. Are there ways to identify the "causative bug" or "category of causative bug" in their participants? Could this be somehow gleaned (even crudely) with the CAP and FP distinctions? The authors do not overreach in their speculations (and this is good) but some of the available NR2F6 literature suggests it may be more important certain sepsis subtypes (e.g. bacterial peritonitis) <https://pubmed.ncbi.nlm.nih.gov/28779026/> <https://pubmed.ncbi.nlm.nih.gov/36052079/> . . . this may be a direction for future inquiry.

1.2 We thank the reviewer for highlighting this interesting potential biological association with bacterial sepsis/peritonitis, and also for their appreciation of the necessity of not speculating too much about specific associations without additional evidence.

We concur that the genetic risk variants we identified could be specific to more homogeneous subtypes of sepsis, and indeed have previously described such effects in our GWAS for mortality in adult sepsis due to CAP (Rautanen et al). Unfortunately, in over half of patients no organism was identified on microbiological culture, so we cannot optimally perform subgroup analyses restricted to bacterial sepsis patients. As noted in the paper referenced by the reviewer (35835848), such subgroup analyses are very difficult to interpret given the limitations in pathogen identification in sepsis. We find that the SRS risk variant associated with *NR2F6* expression is no longer associated with SRS1ever status when the cohort is restricted to FP (bacterial peritonitis) patients, but this could simply be due to the greatly reduced power.

We have added to our discussion around this association to highlight the prior findings relating to *NR2F6*, and to comment on the potential relevance of the infectious agent in identifying risk factors.

“For example, one locus involves an eSNP for *NR2F6*, a transcription factor that represses transcription of key cytokines in CD4⁺ and CD8⁺ effector T cells including IL-2, IFN γ , and TNF α ^{61,62}. The eSNP is located within an ATF3 ChIP peak⁶³ and therefore may be specifically relevant in the context of stress and immune regulation. **The causative pathogen and site of infection are key sources of variation in the sepsis response, and it is likely that there are genetic risk factors specific to particular types of infection. Previous reports raise the possibility that this *NR2F6* association is particularly relevant to bacterial peritonitis (PMID: 35835848, 28779026, 36052079), but our limited cohort size and microbiological information preclude subgroup analyses. With methodological advances, such investigations will be an important direction for future inquiry”**

3) While not necessarily required for this manuscript, two additional contexts may deserve mention as future directions.

The first is the relative timing of SRS1 in these patients. There are many ways to parse this out but the subsections of "ever SRS1" might be important (do temporal trajectories of SRS1 development in relation to other clinical features predict outcomes).

1.3a We agree with the reviewer that the dynamics of SRS status over time is very important to consider.

We are somewhat limited in this cohort by the sampling window used, but we find that it is extremely rare for a patient to move from non-SRS1 to SRS1. Our categorisation of SRS1ever vs never is therefore similar to “SRS status at the first available time point”. Similarly, in our previous paper (Cano-Gamez et al, *Sci Trans Med*), we explored change in the quantitative SRS score (SRSq) within patients over time, and found that 1) in 80% of patients SRSq decreased over time, and 2) those patients where there was negligible or no decrease in SRSq had a higher risk of mortality. However, this was limited to a relatively small number of patients with serial samples and clearly could be explored further in additional datasets, so we have added this point to the discussion:

“The lack of availability of both genotyping and gene expression on the same sepsis patients also precluded validation in external cohorts. Our cohort is restricted to the two main sources of sepsis in the UK (CAP and FP). Whilst we observed a small number of eQTL effects differing by source of infection, potentially driven by location and/or pathogen type, we were not able to investigate pathogen specific effects due to insufficient microbiological information. Our cohort

is of predominantly European ancestry and therefore population specific effects will not be identified. The ideal cohort design to investigate sepsis dependent eQTL effects would include pre-sepsis samples from the same individuals to allow a within-study interaction model. Given this is not feasible, we leveraged publicly available summary statistics^{40,41} to compare effect sizes⁴² between sepsis and health. However, technical differences between studies, such as cohort ancestry, sample size, experimental platform, the set of variants assessed and the availability of summary statistics can confound this comparison. We therefore minimised the impact of these variables by matching them to our cohort as closely as possible. **Additionally, given the potential utility of SRS trajectories over time for prognostication and understanding treatment responses, more comprehensive serial sampling should be employed in future studies.**"

The second piece of context is biological sex. The genetics, physiology, and incidence of sepsis are known to differ between the sexes. <https://pubmed.ncbi.nlm.nih.gov/24966193/> <https://pubmed.ncbi.nlm.nih.gov/35835848/> <https://pubmed.ncbi.nlm.nih.gov/28963611/> <https://pubmed.ncbi.nlm.nih.gov/19996327/> <https://pubmed.ncbi.nlm.nih.gov/14223226/> In short, studies of sepsis should be stratified by sex when possible. Since so much work has already been presented here, the authors could avoid doing this, by simply noting that this should be done in future work.

1.3b We agree that sex is an important factor to consider in understanding heterogeneity in the sepsis response. As mentioned above [response 1.2], we are hesitant to stratify our cohort due to the limited sample size, but we included sex as a covariate in our heritability and GWAS analyses. We note that there is a non-zero effect estimated in the former (0.12 [SE 0.032]), indicating that sex indeed has an impact on the SRS phenotype. We have added the full heritability results to the supplementary tables (Supplementary Table 2; subsequent table numbers changed throughout), and mentioned this topic more generally in the discussion section [section also edited in response 1.2].

Source	Estimate	Standard Error
Genetic variance [V(G)]	0.138966	0.069138
Residual variance [Ve]	0.106676	0.067543
Phenotypic variance [Vp]	0.245642	0.011162
V(G)/Vp	0.565726	0.276889
logL	182.189	
logL0	180.055	
LRT	4.269	
df	1	
Pvalue	1.94E-02	
Sample size	997	
Fixed effects	Estimate	Standard error
Population mean	0.084509	0.177485
Age	0.0102	0.00626
Age^2	-0.000081	0.000053
Genotyping PC1	-0.580608	1.090227
Genotyping PC2	0.199891	0.762984
Genotyping PC3	0.167896	0.682518
Genotyping PC4	-0.096222	0.686925
Genotyping PC5	0.209685	0.634264
Genotyping PC6	0.130617	0.64264
Genotyping PC7	-0.280468	0.611016
Sex	0.122251	0.031656

New Supplementary Table 2: Summary of GCTA-GREML analysis showing estimated variance in SRS phenotype explained by common variants.

Results:

“We estimated the heritability³² as 57% ($\pm 28\%$, $p=0.019$, $n=440$ “SRS1ever” vs 557 “SRS1never” patients), supporting the hypothesis that common variants contribute substantially to SRS during critical illness (Supplementary Figure 1, Supplementary Table 2).”

Discussion:

“The causative pathogen and site of infection are key sources of variation in the sepsis response, and it is likely that there are genetic risk factors specific to particular types of infection. Previous reports raise the possibility that this *NR2F6* association is particularly relevant to bacterial peritonitis (PMID: 35835848, 28779026, 36052079), but our limited cohort size and microbiology information precludes such subgroup analyses. **With methodological advances, such investigations will be an important direction for future inquiry, together with consideration of sex-specific risk factors.**”

We have also performed an additional eQTL interaction analysis to investigate the impact of sex on transcriptomic regulation in sepsis, noting that we have only considered autosomal genes and variants in this analysis. As we had not corrected for sex in the baseline eQTL mapping, we only considered SNP-gene pairs where the eQTL effect remained significant with the addition of sex to the model. Interestingly, we found very few significant interaction effects compared to the other analyses presented (source of sepsis, SRS1 status, cell proportions).

We have added this finding to the results and as a Supplementary Table:

Results:

“We tested each of the independently associated lead SNPs for interaction effects with the source of sepsis (CAP or FP, 12,663 SNP-gene pairs with ≥ 2 minor allele homozygotes in each group tested, Methods). We identified 166 significant interaction effects (FDR <0.05), more than expected by chance (permutation p-value <0.01 , Supplementary Figure 7, Supplementary Table 6), of which roughly half ($n=88$) had stronger effects in CAP (Figure 1e). The eGenes involved were enriched for the Reactome term “Biological oxidations” (FDR=0.0033) and were members of a subnetwork connected by hub genes *APP*, *AKT1* and *ABCC1* (Methods). **For comparison and given the known association between sex and infectious disease**[PMID: 24966193, 28963611, 14223226], we similarly tested for autosomal eQTL interactions with sex and found only 9 significant effects (Supplementary Table 9).”

Methods:

“We tested the signal variants from the conditional eQTL analysis for interactions with SRS1 status, diagnosis (CAP vs FP) and cell proportion (monocyte, lymphocyte, or neutrophil), whilst also correcting for the effect of any independent signal variants. **For comparison, we similarly tested for eQTL interactions with sex. As this variable was not included as a covariate in the original eQTL mapping, we first confirmed that the autosomal SNP-gene pair remained significant with sex added to the model, and then tested for an interaction effect.** SNPs were only tested if there were at least two minor allele homozygote individuals in each subgroup or each half of all cell proportion distributions being tested. Interaction p values were calculated using a likelihood ratio test and corrected for multiple testing using the Benjamini-Hochberg method, and significant interactions identified using an FDR threshold of 0.05.”

4) The excel file in the supplement could benefit from better labels (e.g. in table 2 - what is S05, S001, etc.)

1.4 Many thanks for highlighting this; we have ensured that the supplementary tables are now fully described, in particular the Plink outputs mentioned by the reviewer.

Reviewer #2:

eQTLs identify regulatory networks and drivers of variation in the individual response to sepsis

In this manuscript, Burnham and colleagues discuss eQTL mapping in a cohort of sepsis patients. The paper is very well written and statistically rigorous, leading to exciting and trustworthy insights into the disease. I have a couple of comments that could further improve the work, but overall I am very happy with the status already.

Main comments

1) One of the main points I would like to see addressed is the classification of SRS states. I understand this method has been published before, but the current manuscript would benefit from a brief description of the methodology, what data is required to classify a person, and the characteristics of SRS1 and SRS2 (both mentioned in the text, in the context of a 'spectrum'). Lastly, it would be good to explain how the SRSq score relates to these sub types.

2.1 We have expanded our introduction to more comprehensively introduce and describe SRS and SRSq as follows:

Introduction:

“For example, we have previously shown that Sepsis Response Signature (SRS) subgroups resolve the majority of transcriptomic variation in sepsis⁷ even accounting for different infection sources^{12–14}. We find that SRS1 consistently identifies patients who have an immunocompromised gene expression profile and higher mortality rates, compared to the relatively immunocompetent SRS2, with evidence this is driven by underlying neutrophil dysfunction and altered granulopoiesis. We have developed a machine learning framework (Sepstratifier) to assign SRS status based on the expression of a small set of marker genes, which can be measured with a range of technologies. Acknowledging that these subgroups likely capture different ranges of a continuously varying trait, we additionally developed a quantitative SRS score, SRSq, which encompasses a spectrum from health (values close to 0) through SRS2 to SRS1 (values close to 1). However, the regulatory determinants and predisposing factors for the high-risk SRS1 state are unclear.”

Methods:

“SRS assignment

All patient gene expression samples had Sepsis Response Signature (SRS) assignments as described in Cano-Gamez et al¹³. Briefly, the Sepstratifier R package includes a reference dataset with SRS assignments from the original clustering analysis (Davenport et al 2016, Burnham et al 2017) and uses the expression of 7 pre-selected marker genes to assign SRS membership to new samples. New gene expression data are aligned to the GAINs reference set

using the k-nearest neighbours algorithm, then random forest models are employed to predict SRS and SRSq. We filtered our genotyping cohort to 997 patients with SRS assignments on at least 1 time point from any of microarray, RNA-seq, and qPCR. Where SRS1 status was available from multiple assays on the same time point, we found discrepant assignments for 10 samples. In these cases, we used SRS1 status from RNA-seq preferentially followed by microarray, resulting in 516 SRS1 samples and 818 non-SRS1 samples. We categorised patients as “Ever SRS1” if one or more time points (n1=736 patients, n2=185, n3=76) were assigned to SRS1. Our final cohort comprised 440 ever SRS1 patients and 557 never SRS1.”

2) Second, the authors show a pretty good replication rates with a previous, smaller cohort. However, the other cohort shares almost half of its samples with the current discovery cohort. In order to get a more accurate estimate of replication, the authors should repeat this analysis without the sample overlap, by removing the shared samples from discovery or replication.

2.2 We agree completely with the reviewer that a replication analysis should not include overlapping samples. We had originally presented the correlation analysis including the replicate samples simply to demonstrate the substantial gain achieved with the new RNA-seq cohort, but we appreciate that this can be achieved whilst also performing a true replication analysis. We have therefore removed the 134 repeated samples from the RNA-seq cohort and recalculated eQTL effects using the remaining 689 new samples. There was a small reduction in the number of significant eQTL, likely due to the reduction in power, but the correlation remained high at 0.70 (vs 0.72 in the original analysis).

We have edited the results section in question as follows:

“We have previously described eQTLs in a smaller microarray dataset of patients with sepsis due to CAP (n=240, of which 134 individuals overlap with the RNA-seq cohort, Methods)⁷. After restricting the new RNA-seq dataset to the 689 non-overlapping samples, we find high correlation of eQTL effect sizes with those previously reported across SNP-gene pairs assayed in both datasets (Pearson’s $r=0.70$, Supplementary Figure 6, Supplementary Table 7). However, in the full RNA-seq cohort, ~~The previously reported eQTL effect sizes for SNP-gene pairs assayed in both datasets are significantly correlated with this new dataset (Pearson’s $r=0.72$, Supplementary Figure 6, Supplementary Table 5),~~ but we also identify more than twice the number of eGenes, likely due to the larger sample size and the greater sensitivity of RNA-seq for detecting expression.”

Updated Figure S6: Replication of eQTL results from a microarray sepsis cohort.

Comparison of beta values for all SNP-gene pairs with nominal significance in our previous microarray eQTL study⁷ that were also tested in this RNA-seq cohort. **RNA-seq statistics calculated using the subset of non-overlapping samples.** Blue colour indicates significance in the current study.

3) Last, regarding the Transcription factor enrichment analysis: "We restricted the TF motifs considered to those annotated by Jaspasr 2014 (JASPAR2014 R package112) as being defined in Homo sapiens (n=127/205 included in SNP2TFBS)." Is it correct that you then only used 127 TF motifs for this analysis? Jaspasr has had many updates since 2014, including recently to 2024. I would like to request that you use a more up to date motif set, since the database changes quite a lot over time. Alternatively, you could check for the TF motifs that you've included, how well they align with the more recent versions of the database. The reason I'm asking, is because it is important to have current motif information if you want the SNP to TFBS analysis to work effectively.

2.3 Yes, it is correct that we only used the 127 TF motifs included in the SNP2TFBS tool and annotated as Homo sapiens motifs in JASPAR 2014. We chose to use the SNP2TFBS tool due to its specific consideration of the impact of common variants on TF binding sites, and the ability to find motifs not detected in the reference genome sequenced but introduced by the alternate allele. However, as noted by the reviewer it is limited by its use of the older Jaspasr2014 motif set.

As suggested by the reviewer we have checked whether the 127 motifs we tested were still included in the most recent (2024) JASPAR release, and found this was the case for 124/127 motifs. We therefore removed the three motifs that have been dropped between these two releases (MZF1_5-13, BRCA1,

SMAD2_SMAD3_SMAD4) from our motif set and re-ran the transcription factor prioritisation. Previously, all three motifs had been enriched, with SMAD3 and SMAD4 also having differential inferred activity between SRS groups. We therefore edited the relevant results and methods sections as follows, together with the figures and tables presenting these results:

Results:

“We therefore first identified instances where each of 124 human transcription factor binding motifs were interrupted or introduced by sepsis eSNPs (or their LD proxies) using SNP2TFBS^{47,48} (Figure 3a). We found this was more common when the eQTL had an SRS interaction (median 5 vs 4 binding sites introduced/interrupted per eGene, Wilcoxon p-value= 1.08×10^{-6}). For each TF motif, we then classified eGenes as having at least one or no binding sites altered. We found 56 TF motifs were significantly enriched for alteration among SRS interaction eGenes (Figure 3b, Supplementary Table 13), with the HIF1A-ARNT motif having greatest enrichment. This was significantly more motifs than expected by chance ($p < 0.001$), as computed by permuting eGene interaction status 1000 times and repeating the enrichment analysis (Supplementary Figure 12). [...]

Of the TFs with differential activity, 43 were also enriched in the SNP2TFBS analysis (40 motifs), indicating that they could be driving the differences in the regulatory landscape between SRS groups (Figure 3d).

[...] (Figure 3e, Supplementary Table 13)”

Methods:

“Given SNP2TFBS used an older version of the JASPAR motif set, we checked alignment of motifs with the JASPAR 2024 database and removed 3 motifs that were not present in the latter (n=124/205 motifs included in SNP2TFBS retained).”

Updated Figure 3: Identification of putative driver transcription factors for SRS from eQTL interactions.

We acknowledge that this approach precludes the identification of additional potentially relevant motifs that were not included in the SNP2TFBS database; however, with this additional filtering step it should not be leading to spurious associations with motifs that have since been removed from JASPAR. We have included this as an additional limitation in the discussion:

“Finally, there is currently very limited *in vivo* binding data on different transcription factors in primary cells, particularly in the disease context, and any single TF may have both activating and repressive effects across different genes and contexts. We have therefore used curated regulon and predicted binding sites to prioritise candidate regulatory drivers that could be suitable for longer term functional follow up. **With this approach, we are limited to the factors included in each tool so we may have missed an opportunity to discover additional relevant factors.**”

Minor comments

4) - While the authors explain that the eQTLs are identified in patients, but most of them can be found in blood from healthy individuals as well, they do describe the eQTLs they found as 'sepsis eQTLs' initially. I think it's fairer to use something like 'eQTLs significant in sepsis patients'.

2.4 We are certainly keen to make this distinction clear, and so have edited this and related phrasing throughout as suggested:

Results:

“Five ~~sepsis~~ eQTLs **significant in sepsis** involved SRS1 GWAS SNPs (Supplementary Table 3).”

“Given this evidence, we further quantified this context dependency by comparing ~~sepsis~~ eQTL effect sizes **significant in sepsis** to GTEx”

“We therefore first identified instances where each of 124 human transcription factor binding motifs were interrupted or introduced by ~~sepsis~~ eSNPs **significant in sepsis** (or their LD proxies)”

“we tested for association between each module eigengene and the lead ~~sepsis~~ cis eSNPs identified here”

Discussion:

“Furthermore, through integrating our ~~sepsis~~ eQTL data **from the context of sepsis**”

“As sepsis represents an extreme and systemic response to infection, our ~~sepsis~~ eQTL results **from the context of sepsis** may thus help interpretation of risk variants for a broad range of immune and inflammatory diseases.”

“Additionally, we found that nearly 2000 ~~sepsis~~ eQTL signals **significant in sepsis** had a significant interaction with SRS”

Figure legends:

1f: “Each point represents a lead SNP-eGene pair from the first pass ~~sepsis~~ eQTL mapping **in sepsis patients**”

4d and f: “the lead ~~sepsis~~ eSNPs associated with the ME”

S8: “**Comparison of ~~sepsis~~ eQTLs significant in sepsis to GTEx and eQTLgen results.** Comparison of z-scores for lead ~~sepsis~~ SNP-eGene pairs **significant in sepsis**”

Tables:

S4: “~~Sepsis~~ cis-eQTLs (initial pass) **in sepsis**”

S5: “Conditional ~~sepsis~~ cis-eQTLs **in sepsis**”

Methods:

“Briefly, we used a random 5% subset of the SNP-gene pairs tested in both GAINs and GTEx (“random” tests) to learn the correlation structure across null tests and the lead SNPs for **sepsis** **each significant eGene identified in sepsis** (“strong” tests)”

“We used the SNP2TFBS^{50,51} database to identify instances of **sepsis** eSNPs **significant in sepsis** introducing or interrupting predicted transcription factor binding sites”

“We tested the 12,335 unique signal eSNPs from our conditional **sepsis** eQTLs”

5) - Population(s) in cohort is predominantly European: did you observe any outliers? How did you deal with them?

2.5 Yes, the cohort is predominantly European and the few non-European samples generally showed the expected separation on genotyping PCs but were not outliers in the gene expression dataset. We employed stringent genotyping filtering such that rare variants only present in non-Europeans would not be tested. We additionally included 7 genotyping PCs in our models to correct for population structure, and inspected individual eQTL and interaction boxplots to confirm that there were no systematic batch effects. We have added a description of this observation to the Methods section, together with an additional supplementary figure showing the PCA.

Methods:

“Projecting the samples onto a PCA of 1KGP using King, we found that the vast majority of data points clustered together with European ancestry individuals (Supplementary Figure S2a). We did not remove any individuals based on this PCA, but calculated genotyping PCs on the combined GAINs genotyping data set using Plink⁸⁸ and SNPs with MAF >1% and included genetic PCs in downstream analyses.”

New **Figure S2: Genome-wide association study for SRS1ever vs SRS1never.**

a) Samples were projected into the principal component space of samples from the 1000 Genomes Project.

6) - Sometimes use magnifying, sometimes enhanced. I think would be clearer to use same terminology

2.6 In the original manuscript, we had used “enhanced” when comparing eQTL effects across datasets i.e. sepsis vs health, and “magnified” for within-dataset interactions e.g. SRS1 vs non-SRS1. For clarity, we have changed all instances of “enhanced” to “magnifying”:

Results:

“Specifically, we identified 854 signals with bigger effects and/or only significant in sepsis (“sepsis-magnified”)

“However, sepsis-magnified eQTLs differ significantly from those shared with GTEx”

“We noted that sepsis-magnified eGenes were significantly enriched for SRS interactions”

“This is illustrated by a subset of sepsis-magnified eGenes where the effect of genotype also increases continuously with SRSq score”

Discussion:

“It is likely that some sepsis-magnified or specific eQTLs derive from these leukocyte subpopulations.”

Figure legends:

2b: “**b**) An exemplar sepsis-magnified eQTL that also has a significant positive interaction with SRS1 status.”

S9: “**Characteristics of sepsis-magnified eQTL variants.** Density plots demonstrating how eSNPs involved in sepsis-magnified eQTLs differ from eSNPs involved in eQTLs with comparable effect sizes to GTEx in terms of (left) MAF, (middle and right) distance to the transcriptional start site (TSS) of the eGene.”

Methods:

“We classed those significant in both GAINs and GTex but with opposite directions of effects as “opposite effect” (n=53), and divided the remainder into those with bigger effects and/or only significant in sepsis (“sepsis-magnified”, n=854, 10%) or in HV (“sepsis-dampened”, n=1,272)”

“We used a comparable approach to test for enrichment of gene sets defined in our results (e.g. sepsis-magnified eGenes vs different types of interactions).”

7) - How do you interpret the enrichment of PA in SRS interaction eQTLs?

2.7 PA has been reported to regulate systemic inflammatory responses, and specifically to have a role in mTORC1 activation. We have previously noted the relevance of metabolic changes and glycolysis in SRS1, so it is possible that the phosphatidic acid (PA) pathway and glycerophospholipid metabolism is relevant in this respect. We have included the following in the results:

“eGenes with a magnifying SRS interaction were enriched for the Reactome term “synthesis of PA [phosphatidic acid]” (FDR=0.0069). We have previously noted the relevance of metabolic changes and glycolysis in SRS1, so it is possible that PA is relevant in this respect, given previous evidence of a role in mTORC1 activation and in regulation of systemic inflammation^{46–48}.”

8) - What is FET p-value (p6)?

2.8 Thank you for pointing out this undefined abbreviation, we have edited this in the first instance as follows:

“between SRS groups¹³ (Fisher’s Exact Test [FET] p-value”

9) - Fig 1f: some point in lower right quadrant, that should be opposite direction of effect

2.9 In this figure, we show the results of the comparison of eQTL effects from sepsis to GTEx. There were a small number of eQTL that were significant in sepsis but estimated as non-significant in GTEx by mashr. As noted in the methods, we classed these as “sepsis-enhanced”, regardless of the direction of effect. Those significant in both but with opposite directions of effect were treated separately (yellow points). We believe this strategy makes sense for interpretation of the results, as if the effect is not real (non-significant) the directionality is not meaningful. However, we appreciate that this can lead to confusion as not all points in the top left and bottom right quadrants are labelled “opposite direction of effect”, and so have edited the figure to clarify (sepsis-magnified eQTL non-significant in GTEx now indicated by pale green colour).

NB - the terminology of “sepsis-enhanced” has been edited to “sepsis-magnified” in response to point 2.6.

Updated Figure 1f) Sepsis-dependent eQTL effects identified with mashr. Each point represents a lead SNP-eGene pair from the first pass eQTL mapping in sepsis patients that was also tested for whole blood eQTL in the European subset of GTEx. Posterior effect sizes estimated by mashr

are plotted for GTEx against sepsis, and eQTLs are categorised based on the difference between these estimates. eQTLs significant in sepsis are "shared" if the mashr posterior effect size is in the same direction as and within a factor of 0.5 of the GTEx effect size. Those with bigger or smaller effects in the same direction are "sepsis-magnified" and "sepsis-dampened" respectively. Those significant in both GAINs and GTEx but with opposite directions of effects are "opposite direction of effect". Those only significant in sepsis are also classed "sepsis-magnified", and those significant in neither cohort "not significant".

10) - Fig 2b: add lines per genotype

2.10 To assess and illustrate the relationship between genotype and SRSq for this *FAM89A* eQTL, we fitted an updated linear model as follows (as SRSq had not originally been tested directly):

$$\text{adjusted } FAM89A \text{ expression} \sim SRSq + \text{genotype} + SRSq * \text{genotype}$$

where "adjusted *FAM89A* expression" remains the same as originally plotted (i.e. adjusted for the other covariates included in the original eQTL model).

We have added the fitted lines to the plot and an explanation to the figure legend.

“Updated Figure 2b) An exemplar sepsis-magnified eQTL that also has a significant positive interaction with SRS1 status. Gene expression residuals were modelled with an SRSq-by-genotype interaction to illustrate the continuous relationship of SRS with the genotype effect, with point colour indicating number of copies of the minor allele of rs4378192.”

11) - It would be beneficial to give a bit more info on OCEL1

2.11 The function of the occludin/ELL domain containing 1 (OCEL1) gene is not well understood. In cancer, it has been found to be regulated by mutant p53 (PMID: 22203497) and has been proposed as a

prognostic biomarker (PMID: 32083572). It has also been implicated in Aicardi syndrome through a rare variant (PMID: 26091538). However, these reports do not suggest an obvious mechanism through which OCEL1 could affect the sepsis response. We have acknowledged this uncertainty in the results text:

“Of these, two eQTL signals in the same region (*OCEL1*, a predicted membrane component of unknown function³⁷”

12) - Where do the 288 TFs come from (p7)

2.12 The DoRothEA regulon resource that we used to infer transcription factor activity comprises 288 transcription factors. We have added this information to the results section:

“We therefore calculated transcription factor activity scores in each sample using 288 curated regulons from DoRothEA to pinpoint regulators predicted to vary by SRS (Methods)^{49,50}.”

We have also now noted in the limitations paragraph of our discussion that this part of the analysis is necessarily restricted to regulons that have been curated (see response 2.3).

“We have therefore used curated regulon and predicted binding sites to prioritise candidate regulatory drivers that could be suitable for longer term functional follow up. With this approach, we are limited to the factors included in each tool so we may have missed an opportunity to discover additional relevant factors.”

13) - "suggesting they capture gene sets associated with specific biological processes" à as you've shown that these gene sets are highly correlated to cell counts, I would say this suggests the gene sets are associated with specific cell types, not with processes

2.13 We had intended this phrase to refer back to both the cell marker and pathway enrichment results, so have edited to clarify.

“Furthermore, these modules are enriched for biological pathways and marker genes for more granular blood cell populations^{14,53} suggesting they capture gene sets associated with specific biological processes and/or cell types.”

14) - Do you have a sense of what's happening with module 71? Not enriched for cell markers, only very lightly associated with neutrophils and lymphocytes, but not SRS at all.

2.14 We can't say anything conclusive about this module, though it does seem interesting. It contains 14 zinc finger genes, both activators and repressors, and is enriched for the pathway term “general transcription pathway”. It has a modQTL that passed the sensitivity analysis but did not replicate in the microarray dataset, which is why we had not highlighted it in the main text. It will be interesting to look in additional cohorts to determine whether this is a robust association and highlights the importance of stringent filtering for prioritising associations for follow-up.

15) - Fig 4b: what does 'Outcome' refer to? Is this the survival analysis?

2.15 Yes, this is the survival analysis. We apologise that this was unclear and have changed the label to “Survival”.

Updated Figure 4b.

16) - Fig 4f: there are no genes highlighted in orange, I know you've highlighted the location, but it would be good to include the names SENP7 and IMPG2, and indicate that they are not part of the module

2.16 We have annotated the figure with the cis-eGenes for the modQTL SNPs, making sure it's clear that they are not in this case module member genes.

Updated Figure 4f: Circos plot showing the chromosomal locations of the genes contained in module 47 and the lead eSNPs associated with the ME. Genes that are *cis*-eGenes for these eSNPs are highlighted in orange.

17) - Something weird happened with the font in methods section 'Enrichment and network analysis'.

2.17 We thank the reviewer for spotting this, and have ensured fonts are consistent throughout.

18) - "We assessed eigengene correlation across technologies based on 135 patient samples included in both the microarray and the RNA-seq cohort (Spearman's rho)." à I'm happy that you did this test but I missed it in the result section. What was the correlation?

2.18 We found that the correlation varied by module, with a median absolute rho of 0.77 but ranging from 0.04-0.98. As we showed in Supplementary Figure 18, modQTL replication was associated with higher eigengene correlations, but we have now described this analysis more completely in the results section.

Results:

"We then sought to replicate these modQTL using our previously published microarray gene expression data^{7,12,13} (n=135 samples overlapping RNA-seq, n=506 non-overlapping). We computed module eigengene values for each module gene set with at least 5 genes measured on the microarray. Comparing the module eigengenes across technologies using the 135 overlapping samples, we found in general the correlation was good but varied by module (median |rho| for correlation between RNAseq and microarray module eigengene values: 0.77, range 0.04-0.98). We found that 16/29 lead modQTL replicated in non-overlapping patients, and noted that replicating modQTL had greater correlation between the RNA-seq and microarray module eigengene values computed for the 135 samples included in both datasets (median

[rho] for module eigengenes with replicating modQTL: 0.73; for non-replicating modQTL: 0.49, Supplementary Figure 18).”

19) - modQTL sensitivity analysis: did you remove the cis genes from the conditional lead eSNPs as well?

2.19 Yes, for each modQTL locus we removed all cis-eGenes for which the SNPs associated with the module eigengene were lead eSNPs, including non-primary associations.

We have edited the relevant sentence in the results to make this more explicit, and expanded the methods with the generic example used above:

“We recalculated the module eigengenes excluding all *cis*-eGenes associated with the modQTL SNPs in the conditional eQTL analysis, and retested for association with the lead module SNP.”

Methods:

“We recalculated the module eigengenes in the RNA-seq cohort excluding all conditional *cis*-eGenes for which any associated SNPs were lead eSNPs, and retested for association with the lead module SNP using the same model as above. Significance was determined using the same genome-wide threshold.”

Reviewer #3:

This manuscript reflects a monumental body of work integrating multiple genomic datasets in a well characterized population with sepsis. It addresses numerous important questions in sepsis, building upon the current state of knowledge to ask whether DNA variation influences SRSq, whether this variation is modified by the proportion of neutrophils vs lymphocytes, whether GWAS suggestive SNPs are also eQTLs and whether these seem to be sepsis-dependent, and whether the eQTL seems to vary with the proportion of neutrophils vs lymphocytes. It further generates data about the transcription factors dysregulated in SRS1 / SRSq.

1) In general it is very well written, though dense and assuming a high degree of familiarity with genomic investigations. The portion describing modQTL was a little more difficult to follow; I understood it better after reading the discussion, and I suggest minor editors to more clearly explain this in the results.

3.1 We thank the reviewer for pointing out this section as needing further clarification. We have edited the relevant results section to clarify our approach.

NB See also response to reviewer 2 points 18 and 19.

“Using the WGCNA package^{51,52} we identified 106 co-expression modules, each comprising 11-1,785 genes (Supplementary Table 16, Supplementary Figure 14) with highly correlated gene expression. We summarised the expression of each module with its primary eigengene (the first principal component) to provide a single representative value for the module in each sample. We then correlated these module eigengenes with our features of interest, and found that individual modules were associated with disease phenotypes including SRS, survival and

measured cell proportions (Supplementary Figure 15, Supplementary Table 17). Furthermore, these modules are enriched for biological pathways and marker genes for more granular blood cell populations^{14,53} suggesting they capture gene sets associated with specific biological processes **and/or cell types** (Figure 4a, Supplementary Figure 16, Supplementary Table 18, 19, 20). Finally, to investigate whether these co-expression modules represent sets of co-regulated genes, we tested **the set of genes in** each module for enrichment of known TF targets from DoRothEA⁴⁹ and found at least one significant TF for 43/106 modules (Supplementary Table 21).

[...]

We computed eigengene values for each module gene set with at least 5 genes measured on the microarray. **Comparing the module eigengenes across technologies using the 135 overlapping samples, we found in general the correlation was good but varied by module (median |rho| for correlation between RNAseq and microarray module eigengene values: 0.77, range 0.04-0.98).**

[...]

We recalculated the module eigengenes excluding all *cis*-eGenes associated with the modQTL SNPs **in the conditional eQTL analysis**, and retested for association with the lead module SNP.”

2) The main limitation, which I think could be more clearly discussed, is that due to the availability of the multiple forms of analysis (GWAS + RNAseq), it works exclusively in the population in which SRS and SRSq were already identified, and thus there was no attempt to replicate findings in a new population.

3.2 As noted by this reviewer, a major challenge we face in our approach is the necessity of having both genome-wide genotyping and gene expression data on the same acutely ill sepsis patients. Whilst this presents a challenge in terms of patient recruitment and data availability, we would like to highlight that our recently published R package, SepstratifierR, does facilitate the assignment of SRS outside of the discovery cohort. Encouragingly, it has already been used by us and others to demonstrate that the general features of SRS1 are replicated in other populations (PMID: 36322631, PMID: 37548511, PMID: 37851064, PMID: 37497667); but as the reviewer highlights this replication has been limited to gene expression and clinical data analysis only. Considering aspects of this work that can be replicated with gene expression data only, we find that (as expected given previous replication of gene expression differences) our inferred transcription factor activity differences between SRS groups in general replicate well in the MARS cohort, for which we have previously assigned SRS status (Cano Gamez 2022).

Response Figure 1: Correlation between differential inferred transcription factor activity results in MARS and GAINs. Each point represents one TF, with the effect sizes estimated from linear (mixed) models comparing SRS1 and non-SRS1 samples in MARS (y axis) and GAINs (x axis). Significance in each cohort is indicated by the point colour.

It may be that with COVID or MARS, there would be opportunities to test whether key modQTL or GWAS-eQTL variants were observed in other populations.

We have explored the possibility of conducting such replication analysis in external datasets, but as well as the difficulties with sharing this sensitive data, we would be limited to small cohorts of predominantly European patients with both data types available. As far as we know, genotyping data are not available for the MARS cohort. We accessed the COVID-19 Multi-omic Blood Atlas (COMBAT) data resource, in which SRS status had been previously assigned (Cano-Gamez 2022) giving 32 SRS1ever and 63 SRS1never patients (hospitalised COVID-19 or bacterial sepsis). None of the lead 25 SNPs from our SRS1ever vs never GWAS were significantly associated with SRS1ever status in this dataset, but we believe that this analysis was highly underpowered and so is inconclusive. Power calculations based on the minor allele frequencies and effect sizes we observed in our discovery GWAS indicated that sample sizes of 300-470 patients would be necessary for replication of the SRS1ever associations.

The GAINS population is a somewhat unique sepsis population with lower mortality than many published sepsis genomic cohorts, and it would be appropriate to acknowledge that some features may be unique to this population. However, the authors did appropriately acknowledge the lack of ancestral diversity in GAINS and describe the need for future populations to more thoroughly investigate ancestral-specific findings.

As suggested by the reviewer, we have provided additional details in the discussion on this limitation of the work together with the potential unique features of our specific cohort. We have also added a Supplementary Table summarising the features of the cohort (see response 1.1), which we hope aids interpretation of our results in terms of the population context.

Discussion:

“Although the SRS gene expression signatures have been replicated across cohorts, the lack of availability of both genotyping and gene expression on the same sepsis patients also precluded validation of the genetic associations in external cohorts given the fairly substantial sample sizes that would be required. For example, power calculations indicate sample sizes of 300-470 would be necessary for replication of the SRS GWAS results. Our cohort is restricted to the two main sources of sepsis in the UK (CAP and FP). Whilst we observed a small number of eQTL effects differing by source of infection, potentially driven by location and/or pathogen type, we were not able to investigate pathogen specific effects due to insufficient microbiological information. Our cohort is of predominantly European ancestry and therefore population specific effects will not be identified. Finally, there may be other features unique to this UK cohort, necessitating validation in external datasets. As sampling technologies and study design evolve, this may become more feasible; for example, the use of deferred consent in patient recruitment and rapid high throughput gene expression quantification should enable more representative cohorts to be recruited and profiled both within the UK and internationally.”

Methods:

“The genpwr R package was used to estimate sample sizes required for replication with 80% power, based on the odds ratios and minor allele frequencies of the lead SNPs from the GWAS loci and the case rate observed in this cohort.”

Referees' report, second round of review

Reviewer #1:

Thank you very much for your thoughtful and thorough responses to my comments. I do have one remaining thought but I would not consider it a deal breaker per se.

You note that "... we only considered SNP-gene pairs where the eQTL effect remained significant with the addition of sex to the model." This can be a defensible approach, but it can be argued that conditioning further analysis on the existence of a marginal eQTL effect is not ideal. If there are real effects are in opposite directions in males and females, then they can cancel each other out and yield a non-significant eQTL effect overall. In other words, by requiring a significant eQTL effect prior to looking for interaction, you can remove eQTLs with opposite effects in each sex before you have a chance to evaluate interaction. This could cause real interaction to be missed.

Reviewer #2:

The authors have addressed all my comments in full, thanks

Reviewer #3:

The authors have fully addressed this reviewer's concerns, and it remains an impactful paper.

Authors' response to the second round of review

Reviewer #1: Thank you very much for your thoughtful and thorough responses to my comments. I do have one remaining thought but I would not consider it a deal breaker per se. You note that "... we only considered SNP-gene pairs where the eQTL effect remained significant with the addition of sex to the model." This can be a defensible approach, but it can be argued that conditioning further analysis on the existence of a marginal eQTL effect is not ideal. If there are real effects in opposite directions in males and females, then they can cancel each other out and yield a non-significant eQTL effect overall. In other words, by requiring a significant eQTL effect prior to looking for interaction, you can remove eQTLs with opposite effects in each sex before you have a chance to evaluate interaction. This could cause real interaction to be missed.

We appreciate the reviewer's point around detection of interaction effects. We have generally taken the approach of only testing for interaction effects for signal SNP-gene pairs where an overall eQTL effect was found. In this way, we are confident that genotype is having an impact on gene expression and we greatly reduce our search space for generally weaker interaction effects. This approach has been taken previously in the literature, e.g. PMID: 35100260, PMID: 37805522, PMID: 35972065.

However, we acknowledge that this precludes the detection of complete "crossover" effects across balanced subgroups, such as males and females as described by the reviewer. We think such opposite effects are biologically possible, but are likely quite rare. For example, they have been described across different cell types when eQTL mapping is performed in purified cell populations (e.g. PMID: 22446964 Fig 2B). Given we have mapped eQTL using bulk RNA-seq on a mixed cell population, such effects are likely to be diluted and detectable as a main effect in our cohort according to the predominant cell type.

If we were to test for interaction effects genome-wide, we would gain the ability to detect these balanced opposite effects, but would reduce our overall power dramatically. We have therefore expanded our discussion of the eQTL mapping approach to include acknowledgement of this limitation:

"It has been widely reported that, despite their presumed regulatory activity, GWAS associations only have limited overlap with eQTLs⁷⁵. One possible explanation for this observation is that the majority of eQTL studies have been conducted on healthy individuals, and regulatory variants may only be active in disease-relevant conditions, such as following immune activation. As sepsis represents an extreme and systemic response to infection, our eQTL results in the context of sepsis may help interpretation of risk variants for a broad range of immune and inflammatory diseases. Moreover, elucidating how

environmental context impacts regulatory associations may improve understanding of how genetic variants contribute to complex traits. Additionally, we found that nearly 2,000 eQTL signals significant in sepsis had a significant interaction with SRS, source of sepsis, and/or measured cell proportions. **Given our approach required an eQTL signal in the full cohort prior to interaction testing, interactions with opposite effects may not have been detected due to the average main eQTL effect being non-significant.**

Referees' report, second round of review

NA